# In vivo identification of GTPase interactors by mitochondrial relocalization and proximity biotinylation

Alison K Gillingham[†]*, Jessie Bertram[†], Farida Begum, Sean Munro*

MRC Laboratory of Molecular Biology, Cambridge, United Kingdom

**Abstract** The GTPases of the Ras superfamily regulate cell growth, membrane traffic and the cytoskeleton, and a wide range of diseases are caused by mutations in particular members. They function as switchable landmarks with the active GTP-bound form recruiting to the membrane a specific set of effector proteins. The GTPases are precisely controlled by regulators that promote acquisition of GTP (GEFs) or its hydrolysis to GDP (GAPs). We report here MitoID, a method for identifying effectors and regulators by performing in vivo proximity biotinylation with mitochondrially-localized forms of the GTPases. Applying this to 11 human Rab GTPases identified many known effectors and GAPs, as well as putative novel effectors, with examples of the latter validated for Rab2, Rab5, Rab9 and Rab11. MitoID can also efficiently identify effectors and GAPs of Rho and Ras family GTPases such as Cdc42, RhoA, Rheb, and N-Ras, and can identify GEFs by use of GDP-bound forms.

DOI: https://doi.org/10.7554/eLife.45916.001

*For correspondence:
ag@mrc-lmb.cam.ac.uk (AKG);
sean@mrc-lmb.cam.ac.uk (SM)

[†]These authors contributed equally to this work

Competing interests: The authors declare that no competing interests exist.

## Introduction

The timing and location of many cellular events is controlled by small GTPases of the Ras superfamily (*Takai et al., 2001*). These proteins share a similar nucleotide-binding fold, with the arrangement of surface loops altered depending on whether GTP or GDP is bound (*Vetter and Wittinghofer, 2001*). Although often called GTPases, they typically lack intrinsic GTPase activity and instead nucleotide status is determined by regulatory proteins that either stimulate GTP hydrolysis, or exchange GDP for GTP (*Müller and Goody, 2018*; *Barr and Lambright, 2010*; *Cherfils and Zeghouf, 2013*). This allows the G protein's binding interactions to be regulated and hence they act as molecular switches that generally only bind their effectors when in the GTP-bound state. In this way, upstream regulators can control precisely the location or activity of numerous cytosolic proteins.

The Ras superfamily can be divided into five major families based on sequence and function: the Rab and Arf families that regulate membrane traffic, the Rho family that regulates cytoskeletal dynamics, the Ras family that regulates cell proliferation and differentiation, and Ran that directs nuclear transport (*Rojas et al., 2012*; *Colicelli, 2004*). The large majority of the Ras superfamily GTPases are lipid modified to allow transient attachment to membranes in the active state. The Ras, Rab and Rho families have prenyl groups attached to one or more cysteines at their C-termini, whilst Arfs have a myristoyl group on the N-terminus and Ran is not lipidated. The biological importance of these families is reflected by the fact that their mutation underlies major human diseases. Mutations in the various types of Ras are present in many cancers, and mutations in other members of the superfamily cause disorders of the immune system, the nervous system or of development (*Hutagalung and Novick, 2011*; *Hobbs et al., 2016*; *Maldonado and Dharmawardhane, 2018*).

Understanding the biology of small GTPases requires identifying their upstream regulators and their downstream effectors. Although a lot of progress has been made with some GTPases, the superfamily comprises 167 members in humans and many remain less well understood (*Rojas et al.,*

2012; *Colicelli, 2004*). In addition, new effectors and regulators continue to be identified for even the well-studied members of the family, and it is likely that the repertoire of interactors will vary between different cell types. The two main approaches that have been used for finding effectors have been yeast two-hybrid screens and affinity chromatography (*Gillingham et al., 2014*; *Fukuda et al., 2008*; *Christoforidis et al., 1999a*). Both take advantage of mutations that in many members of the family lock the G protein in an active or inactive conformation (*Vetter and Wittinghofer, 2001*; *Itzen and Goody, 2011*; *Feig, 1999*; *Der et al., 1986*). Although both approaches have had successes they also have limitations. The yeast two-hybrid method relies on both GTPase and effector folding correctly in yeast, and because only a single gene is tested at a time, it precludes detection of effectors that are protein complexes, or at least require associated proteins for stability. The affinity chromatography approach can detect binding to protein complexes, but requires production of correctly folded GTPase, and typically requires large amounts of cell lysate, and interactors have to bind tightly enough to persist during the column washing that is required to remove the rest of the cell lysate before the interactors are eluted and identified.

Recently, proximity biotinylation has emerged as a new approach for identifying protein-protein interactions. If a protein of interest is expressed in cells as a fusion to a promiscuous biotin ligase (BioID) or an engineered peroxidase (APEX) then proteins in the vicinity become biotinylated and can be isolated with streptavidin and identified by mass-spectrometry (*Roux et al., 2012*; *Martell et al., 2012*). However, applying this to small GTPases is potentially problematic since the fusion to the protein that generates biotin must not interfere with effector binding or membrane targeting. In addition, in order to distinguish specific interactors from those biotinylated because they simply happen to be in the same location it is important to compare several different fusion proteins that have a similar location, and yet different GTPases typically act on different membranes making direct comparison more challenging. In addition, the GDP-bound forms that are used as a control in yeast two-hybrid and affinity chromatography are generally not membrane bound and so will have a different background set of bystanders to the GTP-bound form. Here we report a BioID-based approach in which we compare many different small GTPases relocated to the same location - the mitochondrial outer membrane. We find that in this ectopic location there is a similar set of background interactions, but each G protein also interacts with specific effectors and regulators which can be readily identified by comparative analysis.

## Results

### Relocation of Ras family GTPases to mitochondria

To perform BioID with Ras family GTPases we established a system to express them as fusions to the BirA* promiscuous biotin ligase whilst ensuring that they are all at a standard location so as to allow direct comparison (*Figure 1A*). Mitochondria have proven useful for ectopic relocation of cytosolic proteins as there are well defined targeting signals for inserting proteins into the outer membrane of mitochondria without interfering with mitochondrial function (*Silvius et al., 2006*; *Mitoma and Ito, 1992*; *Hoogenraad et al., 2003*). Moreover, proteins directed to mitochondria will have a tightly restricted distribution. Mitochondria have a relatively large area of membrane in which to accommodate an exogenous protein, and by using a mitochondrial transmembrane domain there is little risk of exchange into the cytoplasm. To test this approach we expressed in cells a form of the endosomal Rab protein Rab5A in which the C-terminal four residues that contain the cysteines that receive lipid modification were replaced with the mitochondrial targeting sequence of monoamine oxidase (*Figure 1A*). The BirA* biotin ligase was placed after the C-terminal unstructured linker of the GTPase but before the mitochondrial transmembrane domain, a position close enough to the switch regions to modify effectors without interfering with their binding. Rab5A was expressed with single residue changes that are known to lock it into either the GTP-bound form by blocking GAP activity (Q79L) or the GDP-bound form by interfering with GTP binding (S34N) (*Li and Stahl, 1993*; *Stenmark et al., 1994*).

When expressed in cells the Rab5A chimeras were located on mitochondria, and probing with streptavidin revealed accumulation of biotin on mitochondria indicating that the BirA* part of the chimera had retained activity (*Figure 1B,C*). The presence of Rab5A on mitochondria did not result in the wholesale relocation of endosomes to the mitochondrial surface, and hence biotinylation of

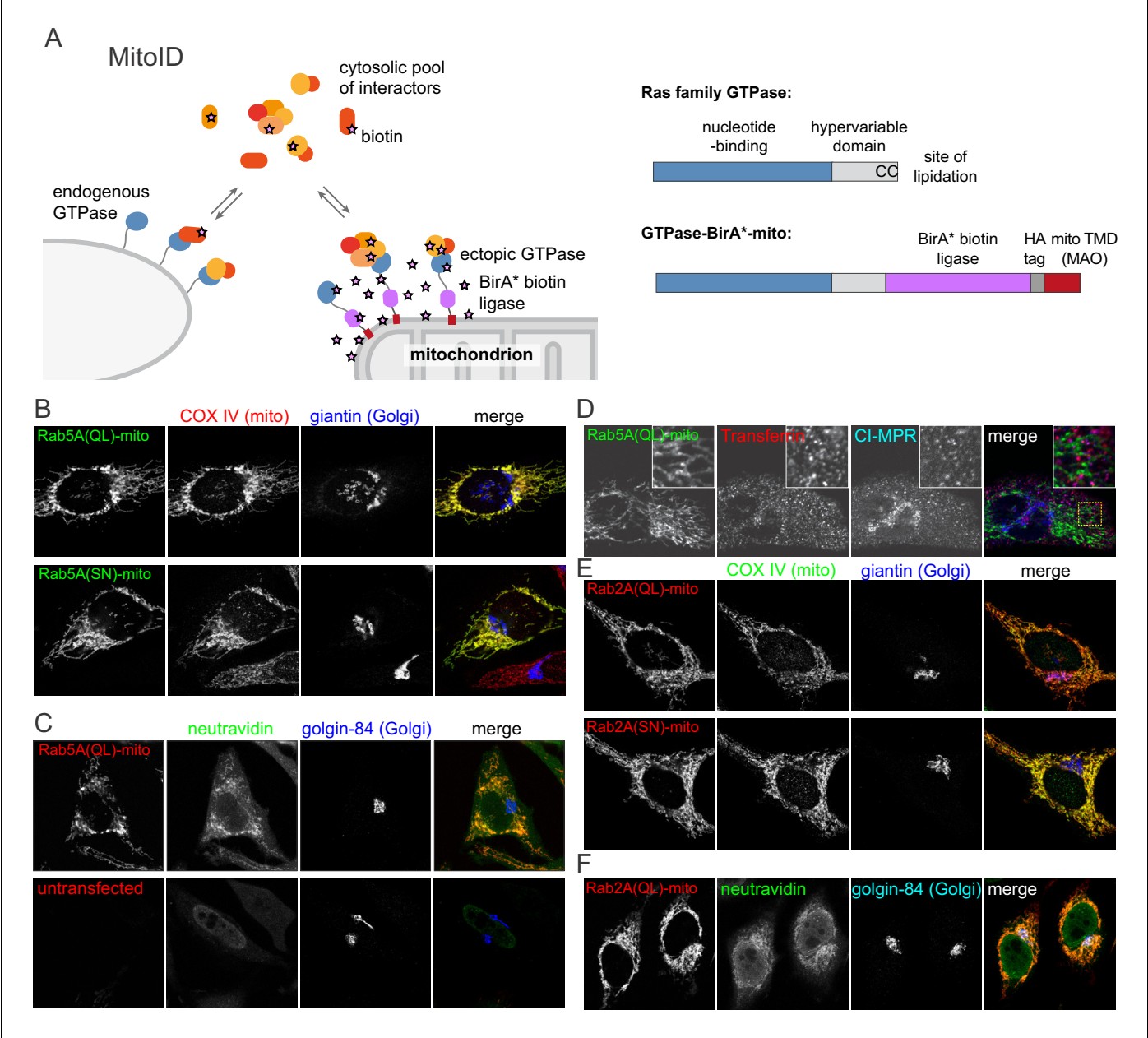

**Figure 1.** MitoID: Expression of BirA*-tagged GTPases on the surface of mitochondria. (**A**) Schematic of the MitoID approach in which a GTPase is expressed as a chimera with BirA* and a mitochondrial targeting sequence. Also shown is the structure of a typical Ras superfamily GTPase and the of chimeric version used for MitoID. (**B**) Confocal images of HeLa cells expressing mitochondrially targeted forms of Rab5A-BirA* as in (**A**), and stained for the HA tag in the chimera as well as markers for the mitochondria and Golgi. The Rab5A chimeras have either the Q79L or the S34N versions that lock them in the GTP- or GDP-bound states respectively. (**C**) Confocal images of HeLa cells expressing mitochondrial Rab5A(Q79L)-BirA*, or untransfected, and labeled for the chimera and with biotin-binding neutravidin and a Golgi marker. (**D**) As (**C**) except that the cells were incubated with fluorescent transferrin for 45 min at 37˚C prior to fixation to label endosomes, and then also stained for cation-independent mannos-6-phosphate receptor (CI-MPR) that recycles through endosomes. Neither endosomal marker is relocated to the Rab5A(Q69L)-covered mitochondria. (**E, F**). Confocal micrographs of cells expressing mitochondrial forms of Rab2A-BirA* in either the GTP-form (Q65L) or the GDP-form (S20N) and stained for mitochondria, the Golgi or with neutravidin as indicated. The Rab2A chimeras accumulate on mitochondria along with biotin, but not the Golgi markers.

DOI: https://doi.org/10.7554/eLife.45916.002

The following figure supplements are available for figure 1:

**Figure supplement 1.** Expression of representative BirA*-tagged GTPases on the surface on mitochondria.

DOI: https://doi.org/10.7554/eLife.45916.003

**Figure supplement 2.** Expression levels of BirA*-tagged GTPases on mitochondria.

*Figure 1 continued*

DOI: https://doi.org/10.7554/eLife.45916.004

**Figure supplement 3.** Expression of BirA*-tagged GTPases on mitochondria does not induce mitochondrial stress.

DOI: https://doi.org/10.7554/eLife.45916.005

endosomal proteins should reflect an interaction with Rab5 and not an altered location of their organelle of residence (*Figure 1D* and *Figure 1—figure supplement 1A*). We also tested a Rab from a different location, the Golgi-localized Rab, Rab2A, and again found mitochondrial accumulation of the chimera, but no gross relocation of the Golgi itself (*Figure 1E,F*).

## Biotinylation of proteins by mitochondrial Rab5A-BirA* chimeras

To determine if the Rab5A-BirA* chimeras biotinylated known interacting proteins, biotinylated proteins were isolated from cells expressing the GTP-bound or GDP-bound forms, and identified by mass-spectrometry. The experiment was performed in triplicate, and to visualize the outcome the mean of the total spectral counts obtained for each protein with the GTP-bound form (Rab5A (Q79L)) was plotted against the same for the GDP-bound form (Rab5A(S34N)). As expected, the highest counts were obtained from Rab5A itself and endogenous enzymes that contain a biotin prosthetic group, and these, along with abundant cytosolic proteins, were found with both Rab5A forms (*Figure 2A*). Examining the proteins with lower counts revealed that ~ 20 proteins known to bind directly to Rab5-GTP, or to be subunits of complexes that bind to Rab5-GTP, where found to be biotinylated specifically by the GTP-bound form (*Figure 2B*). These include canonical Rab5 effectors such as EEA1, Huntingtin and Rabenosyn-5; as well as subunits of complexes that bind Rab5 such as CORVET, class III PI 3-kinase, and FTS/Hook/p107$^{FHIP}$ (*Wandinger-Ness and Zerial, 2014*; *Xu et al., 2008*; *Simonsen et al., 1998*; *Nielsen et al., 2000*; *Christoforidis et al., 1999b*).

We also looked at known regulators of Rab5A. Strikingly, three Rab5 GEFs were present, of which GAPVD1/Rme-6 and ALS2/alsin were predominantly present with the GDP-bound form (*Figure 2B*). The third GEF, Rabex-5, was found at similar levels with both Rab5A mutants, but it is known to form a complex with Rabaptin-5, an effector for Rab5A, an interaction thought to amplify Rab5 activation (*Horiuchi et al., 1997*).

Almost all of these known Rab5 effectors and GEFs were detected in all three of the biological replicates used to calculate the mean values, indicating that the approach is robust and reproducible (*Figure 2C*). Taken together, these results indicate that when Rab5A is fused to BirA* and relocated to the surface of mitochondria it is still able to bind effectors and regulatory proteins in a nucleotide-specific manner. More importantly, these interactions are stable enough for the interaction partners to become biotinylated by BirA* and hence identified by BioID.

## Application of MitoID to a wide range of Ras family GTPases

The above results with Rab5A are a promising indication that the MitoID approach is an effective general means of identifying the binding partners of small GTPases. However, Rab5 is one of the most abundant small GTPases in cultured cells, suggesting that its interactors may also be relatively abundant. Indeed it is one of the few mammalian Rabs for which there is a report of a large scale identification of effectors by affinity chromatography (*Christoforidis et al., 1999a*). To determine if the MitoID approach is broadly effective for small GTPases we applied it to a panel of sixteen other small GTPases. These include ten further Rab GTPases: Rab2A, Rab6A, Rab18, Rab30 and Rab33B from the Golgi, Rab8A and Rab10 from post-Golgi carriers, Rab11A from recycling endosomes and Rab7A and Rab9A from late endosomes (*Hutagalung and Novick, 2011*). In addition, we examined Cdc42, Rac1 and RhoA from the Rho family and Rheb, RalA and N-Ras from the Ras family. All sixteen GTPases were expressed in HEK293 cells as fusions to mitochondrial BirA* with residue changes known, or predicted, to lock them in the GTP-bound or GDP-bound states (*Der et al., 1986*; *Feig, 1999*). Immunofluorescence showed that all of the chimeras accumulated on mitochondria (see *Figure 1—figure supplement 1B* for representative examples). Immunoblotting showed that all were expressed at comparable levels except for the GDP-locked forms of Rab7A, Rab18 and RhoA which appear to be less stable, a phenomenon previously reported for GDP-locked Rab27A (*Figure 1—figure supplement 2*) (*Ramalho et al., 2002*). Finally, we tested whether targeting the

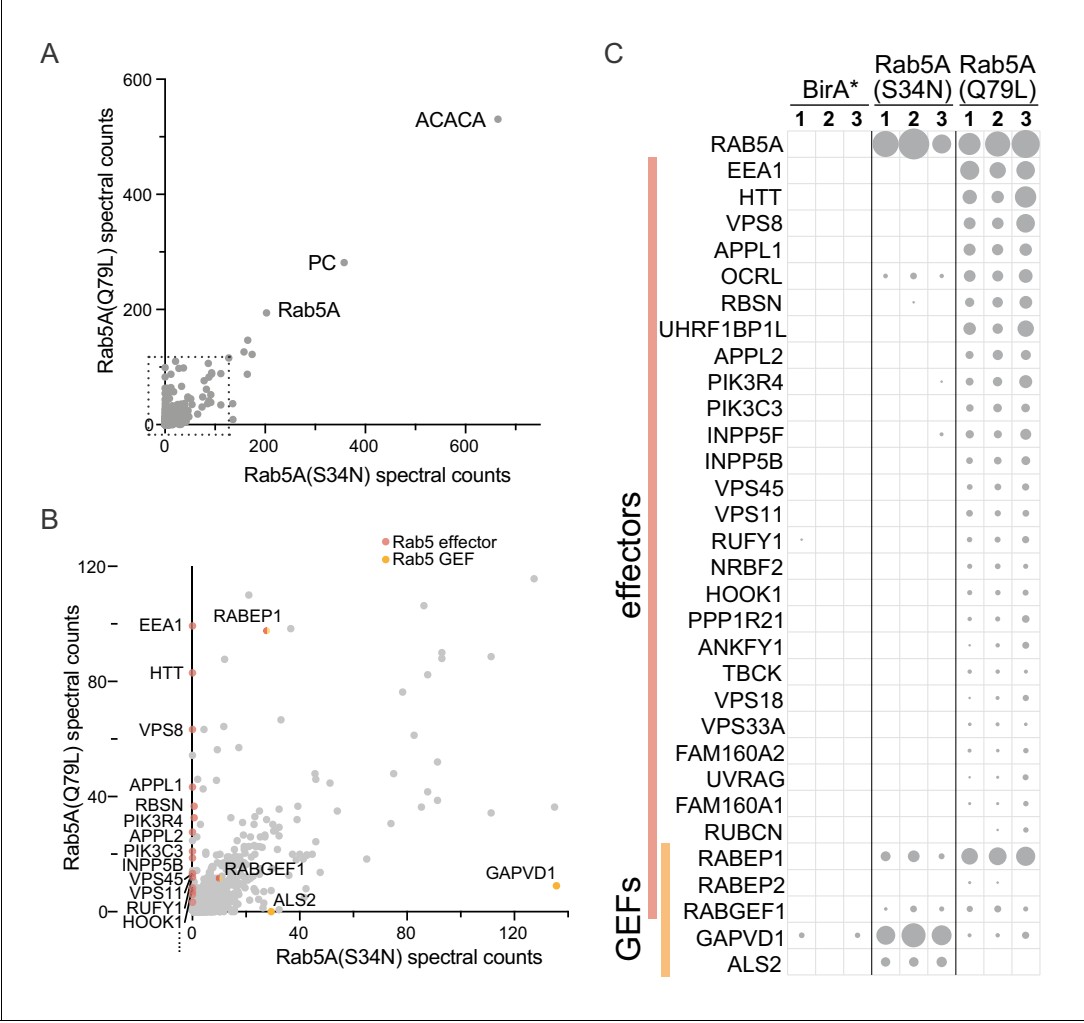

**Figure 2.** Expression of mitochondrial Rab5A-BirA* chimeras leads to biotinylation of known Rab5A effectors and regulators. (**A**) Plot of spectral counts obtained for individual proteins following mass-spectrometric analysis of streptavidin precipitations from cells expressing mitochondrial forms of Rab5A-BirA*. These forms were either GTP-bound (Q79L) or GDP-bound (S34N), and the counts represent means of triplicate biological repeats. The most abundant proteins found with both are Rab5A and the endogenous biotin-containing proteins Acetyl-CoA carboxylase (ACACA) and pyruvate carboxylase (PC). (**B**) As (**A**) except only the region in the dashed box in (**A**) is shown. Known Rab5 effectors and exchange factors are labeled as indicated by their gene names. The effectors are specific for the GTP form (Q79L), whilst the exchange factors are found with the GDP form (S34N). The proteins encoded by RABEP1 and RABGEF1 are found with both forms, but they are known to form a heterodimer that has Rab5 GEF activity but is also a Rab5 effector. (**C**) Spectral counts for known Rab5 interactors obtained with the two mitochondrial forms of Rab5A shown in (**A**) and BirA* alone as a control. Values from three biological replicates are shown, with the area of the circle proportional to the number of counts. For full list of values for all panels see *Supplementary file 1*.

DOI: https://doi.org/10.7554/eLife.45916.006

GTPase chimeras to mitochondria induces mitochondrial stress. The ubiquitin ligase parkin is known to accumulate on stressed mitochondria but no such accumulation was observed in cells expressing mitochondrially targeted GTPases (*Lazarou et al., 2015*) (*Figure 1—figure supplement 3*). Thus, for all sixteen GTPases biotinylation and streptavidin purification was performed as with Rab5A with three biological replicates for each chimera.

## Analysis of MitoID data

Analysis of protein-protein interaction data from conventional coprecipitation experiments has shown that comparing the results from one bait to those from many other baits increases the sensitivity and accuracy with which true interaction partners can be distinguished from non-specific

background (*Huttlin et al., 2015*; *Hein et al., 2015*). Several methods have been developed for such analysis based on either the number of peptide spectra obtained for each protein, or the intensity of the peptide spectra, both of which give an approximate indication of the abundance of a particular protein in a sample. We thus analyzed our data with two methods, one based on spectral counts and other on spectral intensities. The CompPASS platform uses spectral counts and converts these to a 'D-score' by dividing the spectral counts for each protein by the number of baits for which the protein was a hit, and then raising it to the power of the number of times it was replicated with a given bait (*Sowa et al., 2009*). Thus, proteins that are found with all replicates of one bait, but not found with any other bait, have the highest D scores. The second method used spectral intensities and is based on the Perseus platform (*Keilhauer et al., 2015*). The intensities of the peptide spectra obtained with the replicates of one bait are compared to the intensities obtained for all other baits to calculate a fold enrichment and a statistical confidence for that enrichment. The fold enrichment can be plotted against the statistical confidence to give a 'volcano' plot. For both methods consideration has to be given to the likelihood that a real hit with one bait could also be a bona fide interaction partner with some of the other baits which are being used as negative controls as this could reduce the significance assigned. Given that some effectors are known to be shared by several Rabs or several Rhos then this is likely to be the case here. Thus for the D-score we used the WD variant that gives extra weight to interactions where several baits show significantly higher spectral counts than the mean obtained with all the baits (*Behrends et al., 2010*). For the volcano plots based on spectral intensity we did not compare the GTP-bound form of each GTPases to both the GTP and GDP forms of all the other GTPases, but rather we compared each GTP-bound form to all GDP-bound forms on the grounds that effectors generally do not bind to the GDP-form. The GDP-forms were also compared to all other GDP-bound forms as it is unlikely that exchange factors act on more than one or two specific GTPases.

## Application of MitoID to the GTP-bound form of Rab5A

To illustrate the application of WD scores and volcano plots to the analysis of the MitoID data we first consider the GTP-form of Rab5A. As noted above, the MitoID with GTP-bound Rab5A resulted in a large number of proteins that could be precipitated by streptavidin. Ranking of these proteins by total spectral counts showed that as expected many of the proteins were also found with multiple other GTPases (*Figure 3A*). These non-specific interactors include the abundant cytosolic proteins that are typically found as non-specific background in proteomic experiments, along with some mitochondrial proteins (four of the 23 proteins found with every bait). When the WD score method based on spectral counts was applied to the data set it worked very well to enrich known Rab5 effectors with there being nineteen known effectors in the top twenty hits (*Figure 3B*). Likewise, in a volcano plot based on spectral intensities many known Rab5 effectors were amongst those which showed the largest, and most statistically significant, enrichment over background (*Figure 3C*). It is possible to combine the two scorings systems in a 'bubble-volcano' plot, with the area of the data point being proportional to the WD score (*Figure 3D*). The hits with high WD scores are all amongst the most significant hits in the volcano plot indicating a good correlation between the two methods.

In addition to known effectors, there were a number of high scoring hits that have not previously been linked to Rab5. Interestingly, several of these have been reported to act in the endocytic system such as the PX-domain containing kinase RPS6KC1 (*Liu et al., 2005*), the TSC1 and TSC2 subunits of the Rheb GAP (*Manning and Cantley, 2003*), and the V-ATPase regulator WDR7/Rabconnectin-3β (*Sethi et al., 2010*). Other hits include OSBPL9 and its paralog OSBPL11, and the protein RELCH. RELCH was also a hit with Rab11A (*Figure 3B*), and has recently been reported to be Rab11A effector and to have a role in lipid transport (*Sobajima et al., 2018*). To determine if our approach had correctly identified novel Rab5 effectors, OSBPL9 and KIAA1468/RELCH were selected for validation as described below.

These findings show that both methods of analysis correctly identify known Rab5A effectors as the highest hits. The WD-score plot allows an overview of interactions with other GTPases but the bubble-volcano plots are a much more compact means of presenting the hits with a particular Rab. Thus we summarize below the findings with each GTPase using bubble-volcano plots, with the spectral counts, WD scores and peak intensities for all GTPases provided in *Supplementary files 1* and *2*, and the data used for the plots in *Supplementary files 3* and *4*. We recommend using the bubble-volcano plots to get an indication of which proteins are likely to be specific interactors, and

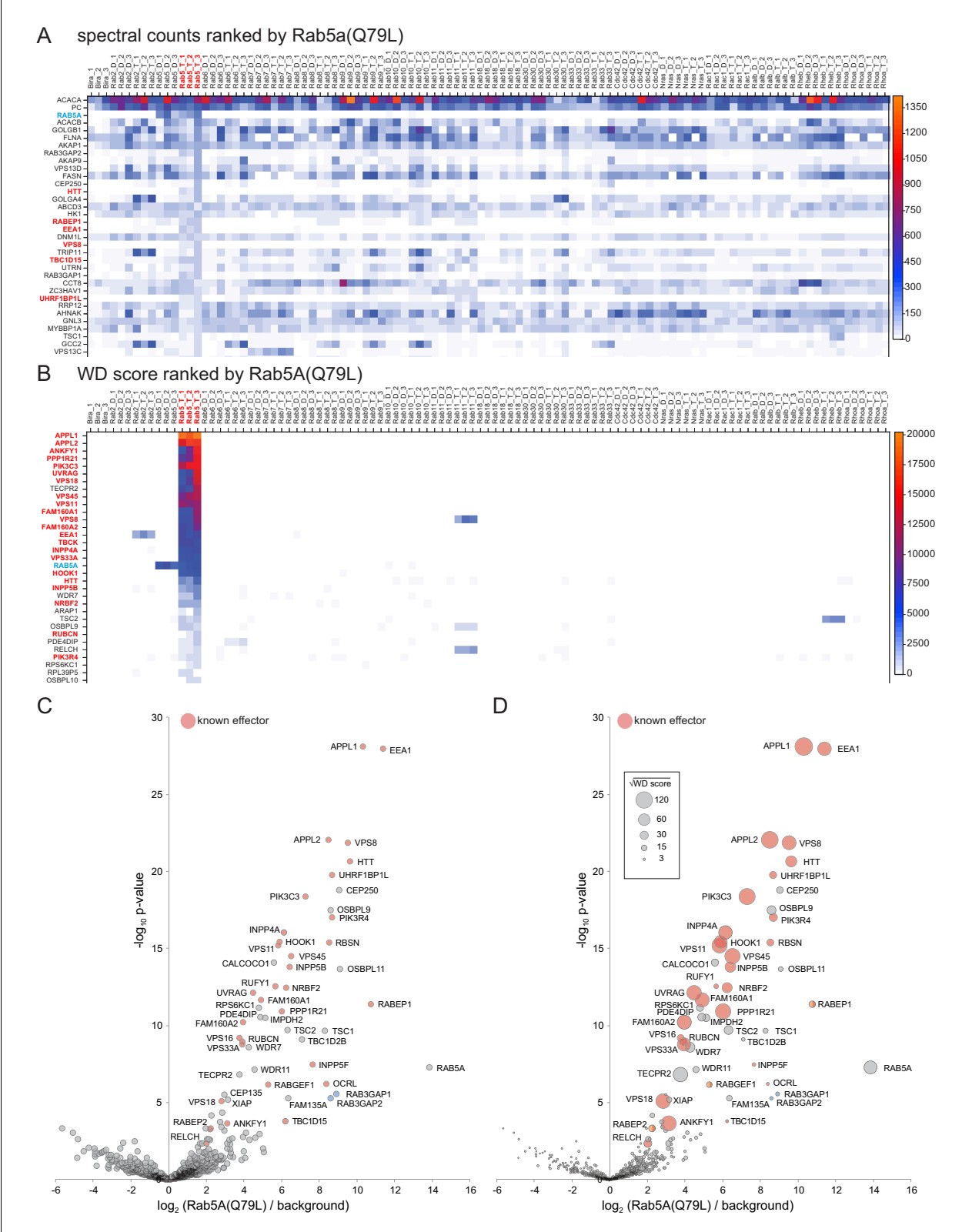

**Figure 3.** Comparison of Rab5A MitoID data to that obtained from sixteen other GTPases. (**A**) Spectral counts for the indicated proteins obtained from triple replicates of MitoID with the indicated GTPases. All GTPases are in either the GDP (**D**) or GTP (**T**) form and the proteins are ranked by their highest spectral counts with the GTP form of Rab5A (Rab5A(Q79L)). Only the top 34 proteins are shown with the full list in *Supplementary file 1*. Known Rab5 effectors are shown red text and Rab5A with blue. (**B**) As (**A**) except that the spectral counts have been converted to WD scores and the
*Figure 3 continued on next page*

*Figure 3 continued*

proteins ranked by their highest WD score with Rab5A(Q79L). Known Rab5 effectors now dominate the highest positions in the list. For the full list of values see *Supplementary file 1*. (C) Volcano plot comparing the spectral intensities from MitoID with Rab5A(Q79L) to MitoID with the GDP-bound forms of Rab5A and sixteen other GTPases (background). Known Rab5 effectors are marked red. Values are in *Supplementary file 3*. (D) Bubble-volcano plot as in (C), but with the area of each point proportional to the root of the WD score. The root was used to ensure that the full range of bubble sizes was visible on the plot.

DOI: https://doi.org/10.7554/eLife.45916.007

The following figure supplement is available for figure 3:

**Figure supplement 1.** Relative distribution of specific and non-specific interactors on bubble-volcano plots.

DOI: https://doi.org/10.7554/eLife.45916.008

examining the table of WD scores to determine if a particular protein binds multiple GTPases. We have also plotted bubble-volcano plots for known effectors of Rab5A-GTP and Rab2A-GTP to allow comparison to the scores for the non-specific interactors obtained with a representative Rab-GDP form for which no plausible interaction partner was found (*Figure 3—figure supplement 1*). This shows that there is a good but not perfect separation, and indicates the area of the plots where real effectors might lie but greater caution is needed. The summaries of the results with Rab GTPases is followed by experimental validation of selected novel hits and then summaries of the results with the Ras and Rho family GTPases.

## Application of MitoID to a panel of Rab GTPases

In addition to Rab5A, we tested ten other members of the Rab family (*Figures 4–6*). We first summarize the findings with the GTP-bound forms, followed by those cases where the GDP-bound form was informative.

### Rab2A

Rab2 is conserved in many eukaryotic phyla and appears to have a dual role on both the Golgi and the endocytic pathway. With Rab2A-GTP the twelve highest scoring hits include four proteins known to bind Rab2 or be in complexes reported to bind Rab2 (*Figure 4A*). These include the coiled-coil protein CCDC186/CCCP-1, a subunit of the COG vesicle tethering complex, and the Bicaudal dynein adaptor (*Gillingham et al., 2014*; *Ailion et al., 2014*). A further seven proteins in the top twelve have been implicated in membrane traffic (e.g. GBF1, RUFY1, WDR11), or are proteins of unknown function that are related to known components of membrane traffic (ARFGEF3).

The next 30 hits include 16 proteins with links to membrane traffic of which six are known Rab2 interactors. The interactors include both subunits of the PICK1:ICA1 complex that acts in secretory granule biogenesis, BLZF1/golgin45, USO1/p115 and INPP5B; as well as the Rab33B GAP TBC1D25/OATL1 that has been reported to have weaker GAP activity to Rab2 (*Itoh et al., 2006*; *Buffa et al., 2008*; *Short et al., 2001*; *Williams et al., 2007*). Of the hits not associated with membrane traffic most are involved with centrosomes and cilia. These may be spurious interactions, but there is evidence that the Golgi is associated with centrioles and with ciliogenesis (*Hua and Ferland, 2018*; *Barker et al., 2016*). The remainder are two proteins of unknown function (STAMBPL1 and FAM184A) and an abundant cytosolic enzyme argininosuccinate synthetase which seems unlikely to be valid. Thus, MitoID identified known effectors of Rab2, and several putative novel effectors, of which ARFGEF3 and STAMBPL1 were selected for validation as described below.

### Rab6A

Rab6 is conserved in most, if not all, eukaryotes and acts on the trans Golgi network, particularly in recycling from endosomes (*Pfeffer, 2011*). Almost half of the top thirty hits with Rab6A-GTP are known Rab6 effectors, including p150/glued subunit of dynactin (DCNT1), the Rab39 GEFs DENDD5A and DENND5B, the dynein adaptor bicaudal-1/2, the Rho GEF ARHGEF1 and the coiled-coil proteins ERC1/2 and TMF1 (*Short et al., 2002*; *Matanis et al., 2002*; *Fukuda et al., 2011*; *Shibata et al., 2016*; *Fridmann-Sirkis et al., 2004*; *Grigoriev et al., 2007*) (*Figure 4B*). In addition, RABGAP1 and its paralog RABGAP1L have been reported to have GAP activity on Rab6 (*Cuif et al., 1999*). There are also several proteins previously linked to membrane traffic at the Golgi and

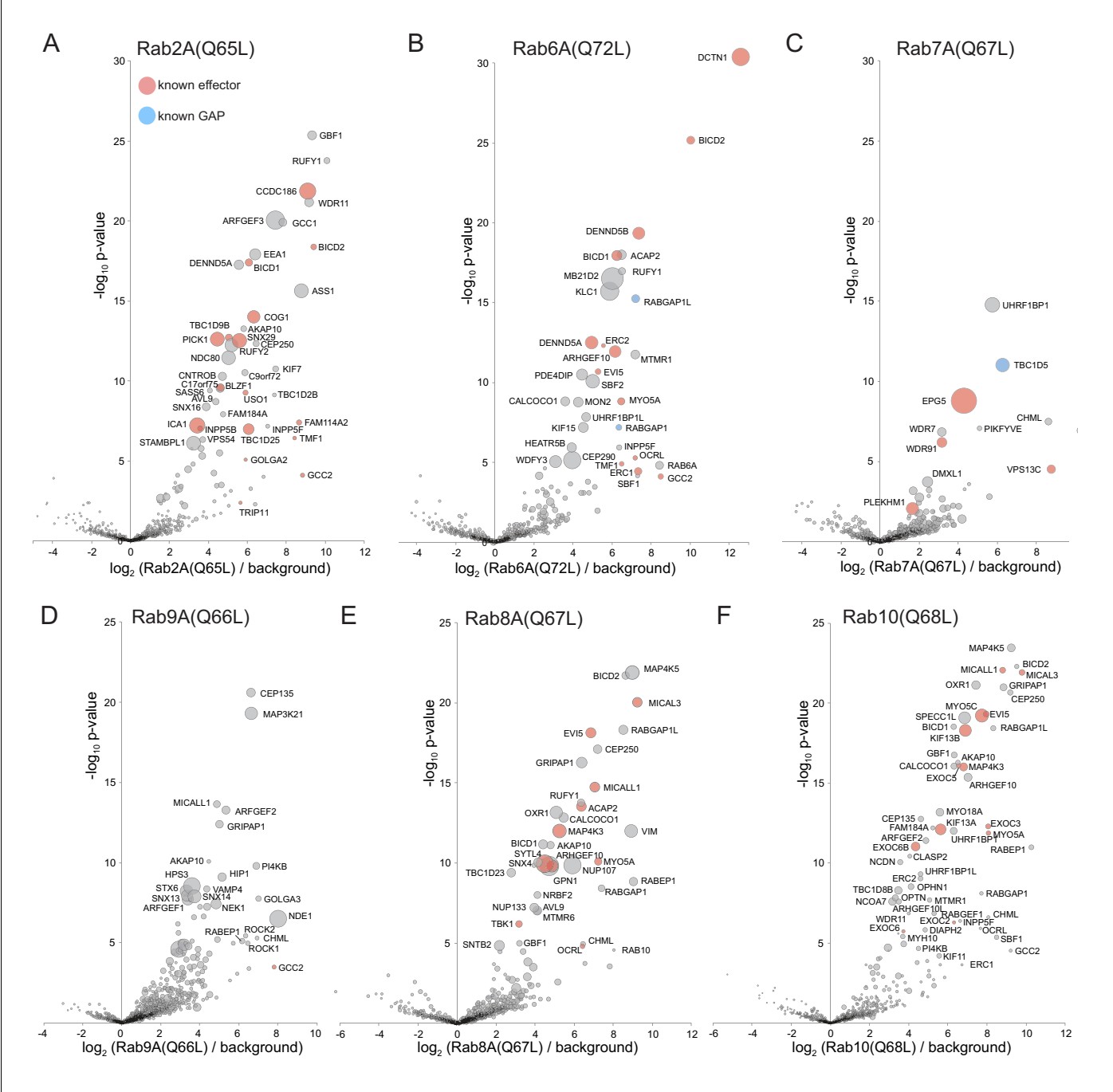

**Figure 4.** Application of MitoID to GTP-locked forms of Rab2A, Rab6A, Rab7A, Rab9A, Rab8A and Rab10. (A–F). Bubble-volcano plots of MitoID with the indicated Rabs, each with a mutation predicted to lock it in the GTP-bound form. In each case the Rab is compared to a background comprising the GDP-locked/empty forms of all seventeen GTPases, with the size of the bubbles proportional to the WD scores. Indicated are known effectors (red) and GAPs (blue). For each plot the Rab is omitted and all values are in *Supplementary file 3*.

DOI: https://doi.org/10.7554/eLife.45916.009

endosomes that are not known Rab6 effectors including the lipid phosphatases MTMR1 and INPP5F, the Beach domain protein WDFY3, the microtubule regulator PDE4DIP/myomegalin, and kinesin light chain; as well as MB21D2, a protein of unknown function.

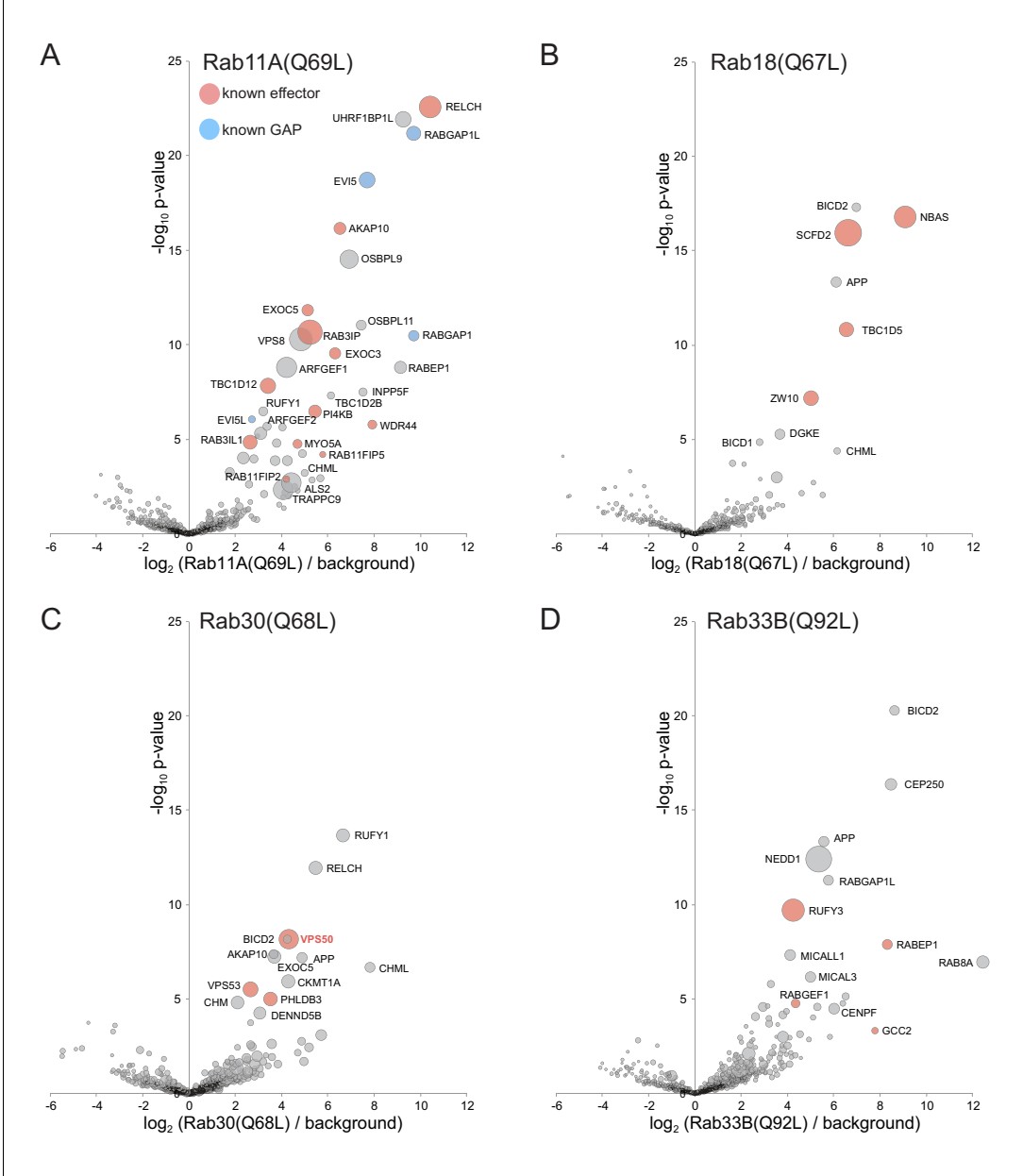

**Figure 5.** Application of MitoID to GTP-locked forms of Rab11A, Rab18, Rab30, and Rab33B. (**A–D**). Bubble-volcano plots of MitoID with the GTP-locked forms of the indicated Rabs. In each case the Rab is compared to a background comprising the GDP-locked forms of all seventeen GTPases, with the size of the bubbles proportional to the WD scores. Indicated are known effectors (red) and GAPs (blue). For each plot the Rab is omitted and all values are in **Supplementary file 3**.

DOI: https://doi.org/10.7554/eLife.45916.010

## Rab7A

Rab7 is conserved in most eukaryotes and acts on late endosomes to control their maturation and fusion with lysosomes. The top hits with Rab7A-GTP include three known effectors: Epg5, Vps13C and WDR91; a known Rab7 GAP, TBC1D5; and the Rab geranyltransferase subunit CHML (**Wang et al., 2016**; **Tabata et al., 2010**; **Casanova and Winckler, 2017**; **Seaman et al., 2009**) (**Figure 4C**). Other strong hits include the PtdIns 5-kinase PIKFYVE which is found on Rab7-containing late endosomes, and WDR7 and DMXL1, which are orthologs of the subunits of the rabconnectin dimer that controls V-ATPase activity in the endosomal system (**Sbrissa et al., 2002**; **Sethi et al.,**

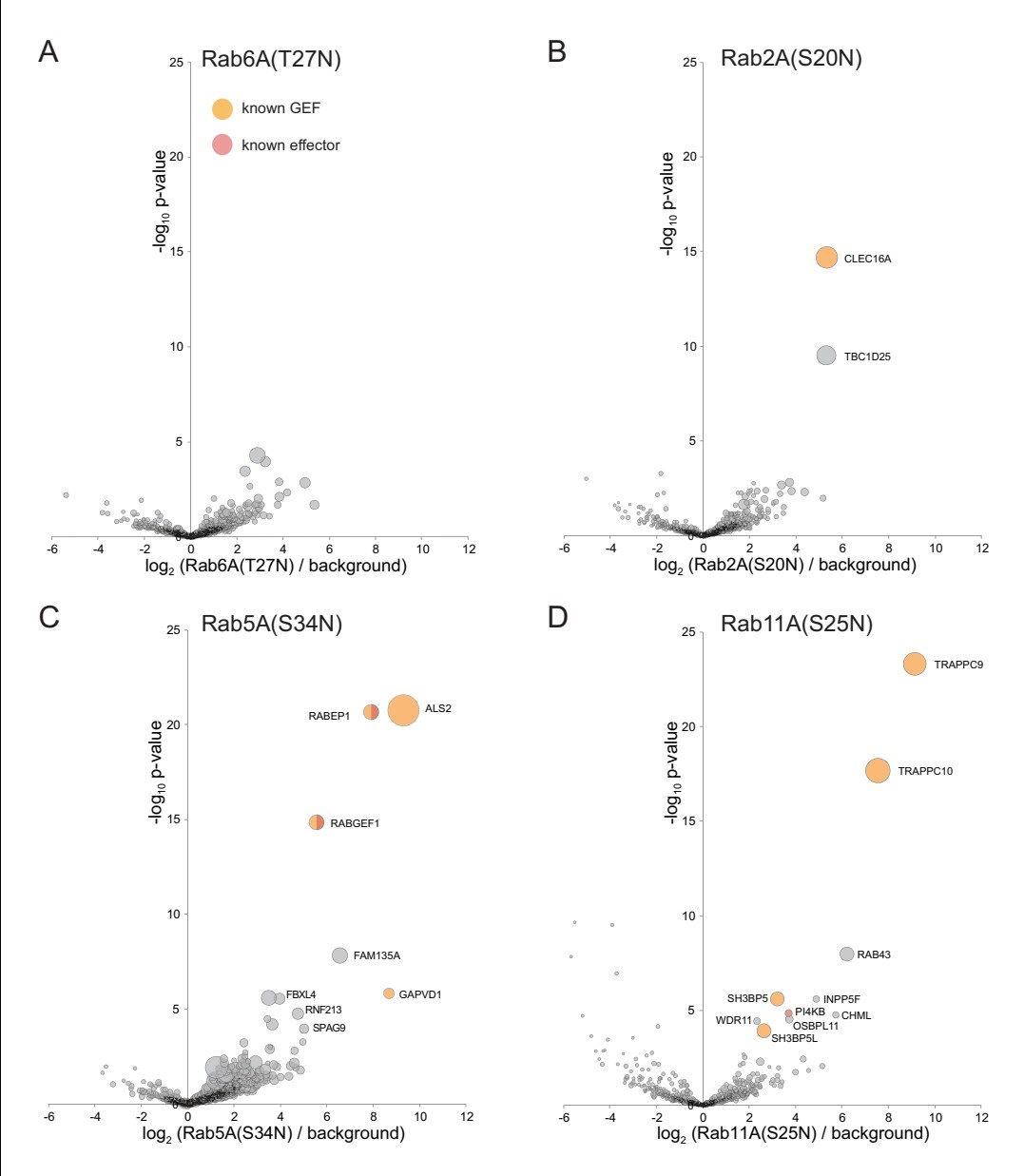

**Figure 6.** Application of MitoID to GDP-locked/empty forms of Rab6A, Rab2A, Rab5A, and Rab11A. (**A–D**). Bubble-volcano plots of MitoID with the indicated Rabs, each with a mutation predicted to lock it in the GDP-bound form. In each case the Rab is compared to a background comprising the GDP-locked forms of seventeen GTPases, with the size of the bubbles proportional to the WD scores. Indicated are known effectors (red) and GEFs (yellow). Note that some of the lower scoring hits were also found with the GTP-bound form of the respective Rabs (eg TBC1D25 for Rab2A, FBXL4 and SPAG9 for Rab5A). In one case (PI4KB with Rab11A) the interaction has been reported to be nucleotide-independent, and so such hits seem more likely to be similar nucleotide-independent effectors rather than GEFs (*Burke et al., 2014*). For each plot the Rab is omitted and all values are in *Supplementary file 3*.

DOI: https://doi.org/10.7554/eLife.45916.011

*2010*; *Yan et al., 2009*). UHRF1BP1 has no reported function, but is a paralog of UHRF1BP1L/ SHIP164 which was reported to be localized to endosomes and to interact with SNARE proteins (*Otto et al., 2010*).

## Rab9A

Rab9A is a relative of Rab7 that emerged early in metazoan evolution and also acts on endosomes. The top WD scores with the GTP form of Rab9A include several proteins known to act on endosomes including the scaffold protein GRIPAP1/GRASP-1; the HPS3 subunit of the BLOC-2 complex that regulates endosomal traffic; the sorting nexin Snx13 and its paralog Snx14; and the monooxygenase MICALL1 (*Gautam et al., 2004*; *Zheng et al., 2006*; *Beer et al., 2018*; *Hoogenraad et al., 2010*; *Sharma et al., 2009*) (*Figure 4D*). Other hits included the known Rab9 effector GCC2/golgin-185 and the dynein regulator NDE1/NudE, which was reported as a hit with Rab9 in a yeast two-hybrid screen but did not show nucleotide-specific binding by co-precipitation with Rab9B (*Bradshaw and Hayashi, 2017*). Of these hits we selected HPS3 and NDE1 for validation as described later.

## Rab8A and Rab10

Rab8 and Rab10 arose from a gene duplication early in metazoan evolution and have related roles in exocytosis including sharing some effectors (*Sato et al., 2014*; *Elias et al., 2012*; *Klöpper et al., 2012*). The high scoring hits with the GTP-bound form of Rab8A include known effectors such as synaptotagmin-like 4 (SYTL4), the myosin MYO5A, the lipid phosphatase OCRL, the Rho GEF ARH-GEF10, and the flavoprotein monooxygenases MICALL1 and MICAL3, with the latter four also being hits for Rab10 (*Fukuda et al., 2002*; *Shibata et al., 2016*; *Sharma et al., 2009*; *Grigoriev et al., 2011*; *Sun et al., 2014*) (*Figure 4E,F*). The Ste20 kinase MAP4K3 and its paralog MAP4K5 were hits with both Rab8A and Rab10. MAP4K3 has recently been reported to co-precipitate with Rab8A, and another paralog, MAP4K2, was reported to bind Rab8 (*Steger et al., 2017*; *Ren et al., 1996*). The other top hits with Rab10 include the known effectors MYO5C and several subunits of the exocyst complex including its direct binding partner EXOC6 (*Sano et al., 2015*; *Roland et al., 2009*). Other strong hits include the motor proteins MYO18A and kinesin-2 KIF3A/B.

## Rab11A

Rab11 is conserved in most eukaryotes and participates in traffic through recycling endosomes and the trans Golgi network (*Wandinger-Ness and Zerial, 2014*). Amongst the top ~30 hits are thirteen known Rab11 effectors: the Rab8 GEF Rab3IP/Rabin8, the Rab GAP TBC1D12, the PI 4-kinase PI4KIIIβ, subunits of the exocyst complex, the A-kinase-anchoring protein D-AKAP2/AKAP10, the kinesins KIF13A/B, and the proteins of unknown function WDR44/Rab11BP and RELCH/KIAA1468 (*Knödler et al., 2010*; *Sobajima et al., 2018*; *Longatti et al., 2012*; *Zeng et al., 1999*; *Eggers et al., 2009*; *Oguchi et al., 2017*; *Etoh and Fukuda, 2019*) (*Figure 5A*). In addition, the hits include two TBC domain GAPs reported to act on Rab11, EVI5 and RABGAP1 (*Fuchs et al., 2007*; *Laflamme et al., 2012*). Interestingly most of the remaining hits are proteins linked to endosome function but not reported as Rab11 effectors. These include the Rac1 and Rab5 GEF ALS2/alsin, the coiled-coil protein RUFY1, the Vps8 subunit of the CORVET complex, and UHRF1BP1L (*Topp et al., 2004*; *Otto et al., 2010*; *Yamamoto et al., 2010*). Of these ALS2/alsin was selected for validation as described below.

## Rab18

Rab18 is conserved in many phyla but its role remains to be fully resolved with evidence for action in both ER to Golgi traffic and in lipid droplet function (*Gerondopoulos et al., 2014*; *Dejgaard et al., 2008*). The top hits with the GTP-bound form included the NAG/NBAS and ZW10 subunits of the ER-associated NAG-RINT1-ZW10 (NRZ) tethering complex, and the Rab7 GAP TBC1D5, with both of these having been previously reported as Rab18 interactors (*Xu et al., 2018*; *Gillingham et al., 2014*) (*Figure 5B*). The other top hits were SCFD2, a poorly characterized relative of the Sec1 SNARE binding protein that has been reported to interact with subunits of the NRZ complex (*Tagaya et al., 2014*; *Huttlin et al., 2015*), and the Bicaudal dynein adaptor which is known to interact with several other Rabs.

## Rab30

Rab30 is a relative of Rab1 that emerged early in metazoan evolution, and is localized to the Golgi although its role is unclear (*Kelly et al., 2012*). Hits with Rab30 included subunits of the GARP

complexes which were also found by affinity chromatography to bind to *Drosophila* Rab30, and PHLDB3, a protein of unknown function found to co-precipitate with Rab30 in a large scale proteomic screen (*Huttlin et al., 2017*; *Gillingham et al., 2014*) (*Figure 5C*).

### Rab33B

Another relative of Rab1, Rab33 also emerged early in metazoan evolution with two paralogs arising with the expansion of vertebrates, both of which contribute to membrane traffic at the Golgi apparatus (*Zheng et al., 1998*). High scoring hits included three previously reported effectors: RUFY3, the RABEP1/RABGEF1 (Rabex5/Rabaptin) complex, and the golgin GCC2/golgin-185 (*Valsdottir et al., 2001*; *Fukuda et al., 2011*; *Hayes et al., 2009*) (*Figure 5D*). Other hits include several centrosomal proteins, the plausibility of which remains unclear, and members of the MICAL family which bind Rab1 and several other Rabs (*Rai et al., 2016*).

## GDP-locked forms of some Rabs can detect exchange factors

Most, if not all, Rabs are activated by GEFs that open the nucleotide-binding pocket and thus release bound GDP and enable binding of GTP. In those cases characterized, the GEF binds more tightly to the nucleotide-free form than the GTP-bound form to allow release of the latter following exchange (*Müller and Goody, 2018*; *Cherfils and Zeghouf, 2013*) (*Langemeyer et al., 2014*; *Koch et al., 2016*; *Feig, 1999*). Thus, mutations which interfere with GTP-binding stabilize the interaction with the GEF, and overexpression can cause a dominant negative suppression of activation of the endogenous Rab. In this study we have used the canonical GDP-locked forms in which the conserved P-loop serine or threonine is replaced with asparagine. This mutation has been shown to prevent GTP binding, but it also reduces affinity for GDP, and hence it is regarded as a hybrid GDP-bound and nucleotide-free form (*Koch et al., 2016*; *John et al., 1993*). In contrast to our findings with the GTP-locked forms, most of the S/T->N forms of the Rabs do not show any highly significant interactions (illustrated for Rab6A(T27N) in *Figure 6A*). However, three Rabs showed strong and reproducible interactions with known or putative exchange factors.

Rab2A(S20N) gave two strong hits (*Figure 6B*). The first is CLEC16A, a protein linked to autophagy, with the *C. elegans* ortholog, GOP-1, reported to be required for Rab2 activation in vivo and in vitro (*Yin et al., 2017*; *Redmann et al., 2016*). The second strong hit is TBC1D25/OATL1 which, as noted above, has been reported to be a Rab2 GAP, suggesting that it can bind both forms of the GTPase prior to GTPase stimulation. With Rab5A(S34N) the five most significant hits include the Rab5 GEFs ALS2/alsin and GAPVD1/Rme-6, and the subunits of the dimer formed by the Rab5 GEF rabex5/RABGEF1 and rabaptin-5/RABEP1 (*Figure 6C*). The remaining protein, FAM135A, has no previously reported function. Finally, with Rab11A(S25N) the two top hits were the C9 and C10 subunits of the TRAPPII complex that activates Rab11 (*Riedel et al., 2018*) (*Figure 6D*). The other two known Rab11 GEFs, SH3BP5 and its paralog SH3BP5L were lower scoring hits, along with PI 4-kinase that binds Rab11 in a nucleotide-independent manner and the Rab geranyltransferase subunit CHML (*Sakaguchi et al., 2015*; *Burke et al., 2014*). OSBPL11 and INPP5F were also lower scoring hits, but were also hits with Rab11A-GTP and so are candidates for proteins that, like PIK4B, bind in a nucleotide independent manner. Taken together, these results show that MitoID can also detect interactions between the GDP-bound/nucleotide free versions of some of the Rabs and their exchange factors, and this was found to work even better for the Ras and Rho family GTPases described later in the paper.

## Validation of novel interaction partners for Rab2A, Rab5A, Rab9A and Rab11A

The GTP-forms of the eleven Rabs analysed here all had highly significant hits that have not previously been reported to be GTPase effectors. To further validate the MitoID approach we tested a subset of these hits.

The highly significant hits for Rab2A-GTP included STAMBPL1 and ARFGEF3. STAMBPL1 is a paralog of STAMBP (AMSH), an endosome-associated deubiquitinase that regulates entry of endocytosed proteins into the ESCRT-dependent multi-vesicular body pathway (*Clague and Urbé, 2006*). Although STAMBPL1 has been shown to share with STAMBP a specificity for cleaving Lys63-linked polyubiquitin chains, its function is unclear (*Sato et al., 2008*). ARFGEF3 (BIG3) is a member of the

Sec7 domain family of Arf GEFs, and regulates insulin granule formation via an unknown mechanism (*Li et al., 2014*). Both STAMBPL1 and ARFGEF3 specifically bound to Rab2A-GTP when cell extracts were applied to GST-Rab-coated beads (*Figure 7A*). In addition, GFP-tagged forms of the proteins showed a striking relocation to mitochondria when co-expressed with the mitochondrial form of Rab2A-GTP (*Figure 7B,C*). Finally, we were able to express recombinant STAMBPL1 and this bound directly to recombinant Rab2A (*Figure 7D*). These findings clearly suggest that Rab2 can recruit STAMBPL1 and ARFGEF3 to membranes, and imply that both proteins are regulated, at least in part, by Rab2. Interestingly there is evidence that ARFGEF3, like Rab2, has a role in secretory granule biogenesis (*Buffa et al., 2008*; *Csizmadia et al., 2018*; *Sumakovic et al., 2009*).

Rab5A-GTP gave several novel hits including OSBPL9, RELCH and TBCK. OSBPL9 (ORP9) is a member of a family of cholesterol transfer proteins that binds to the protein VAP in the ER and to PtdIns(4)P on the Golgi and endosomes (*Ngo and Ridgway, 2009*; *Dong et al., 2016*). RELCH has recently been reported as a Rab11 effector and to have a role in cholesterol transport (*Sobajima et al., 2018*). TBCK has a kinase domain and a TBC domain that is found in many Rab GAPs, but its function is unknown (*Chong et al., 2016*). All three proteins bound specifically to the GTP form of Rab5 when cell extracts were passed over Rab5-coated beads (*Figure 7E*). In addition, a GFP-tagged form of TBCK relocalized to mitochondria when co-overexpressed with mitochondrial Rab5-GTP (*Figure 7F*).

Rab9 is located on late endosomes and has many fewer reported effectors than Rab2 or Rab5, and so it was interesting to find several strong novel hits. We selected for analysis NDE1, a dynein regulator, and the HPS3 subunit of the BLOC-2 complex that has an unresolved role in the biogenesis of lysosome-related organelles (*Dennis et al., 2015*; *Bradshaw and Hayashi, 2017*). For NDE1 it was possible to express the protein in *E. coli*, and this recombinant protein bound directly to the GTP-from of Rab9A (*Figure 7G*). There were no suitable antibodies to investigate HPS3, but when epitope-tagged HPS3 was co-expressed in cells with mitochondrial Rab9A it was efficiently recruited by the GTP form but not the GDP form, and the same tagged proteins showed GTP-specific binding to a Rab9A affinity column (*Figure 7H,I*). Rab9 recruiting to endosomes NDE1 and the BLOC-2 complex would be consistent with NDE1 being required for endosomal positioning and Rab9 having a role in melanosome biogenesis (*Mahanty et al., 2016*; *Lam et al., 2010*). Finally, Rab11A is present on recycling endosomes and the TGN and amongst the novel hits with Rab11A-GTP was the Rac1 and Rab5 GEF ALS2/alsin, an interaction confirmed by affinity chromatography of cell extracts with Rab11A-coated beads (*Figure 7J*). Taken together these results show that the MitoID method is capable of identifying novel effectors of even well studied Rabs.

## Application of MitoID to Ras GTPases

The Rabs are only one of five families within the Ras superfamily of Ras GTPases (*Rojas et al., 2012*; *Takai et al., 2001*; *Colicelli, 2004*). To determine if the MitoID approach was broadly applicable we tested it with representative GTPases from the Rho and Ras families. The Ras GTPases have diverse roles in regulation of cell growth and differentiation (*Hobbs et al., 2016*; *Heard et al., 2014*; *Gentry et al., 2014*). To assess the applicability of MitoID to this family we selected three well characterized members: N-Ras, RalB and Rheb.

### N-Ras

One of several closely related Ras GTPases that act in signaling pathways controlling cell growth and that have received extensive attention due their being oncogenes (*Hobbs et al., 2016*). With the GTP-bound form of N-Ras, the top five hits comprise two Ras effectors (the kinases RAF1/c-Raf and BRAF1/B-Raf), the Ras GAP NF1/neurofibromin-1, and the closely related proteins RAPGEF2 and RAPGEF6 (*Figure 8A*). The latter two proteins act on Rap1, but also have a Ras associating (RA) domain found in other proteins to bind Ras or other small GTPases. RAPGEF2 has been reported to bind Rap1 but there is no report of an interaction with N-Ras (*Rebhun et al., 2000*; *Liao et al., 1999*). With the GDP-form of N-Ras the top hit is the Ras GEF SOS1 (*Figure 8B*), with there also being an interaction with RAP1GDS1 that has a chaperone-like role and binds Ras irrespective of its nucleotide state (*Vikis et al., 2002*).

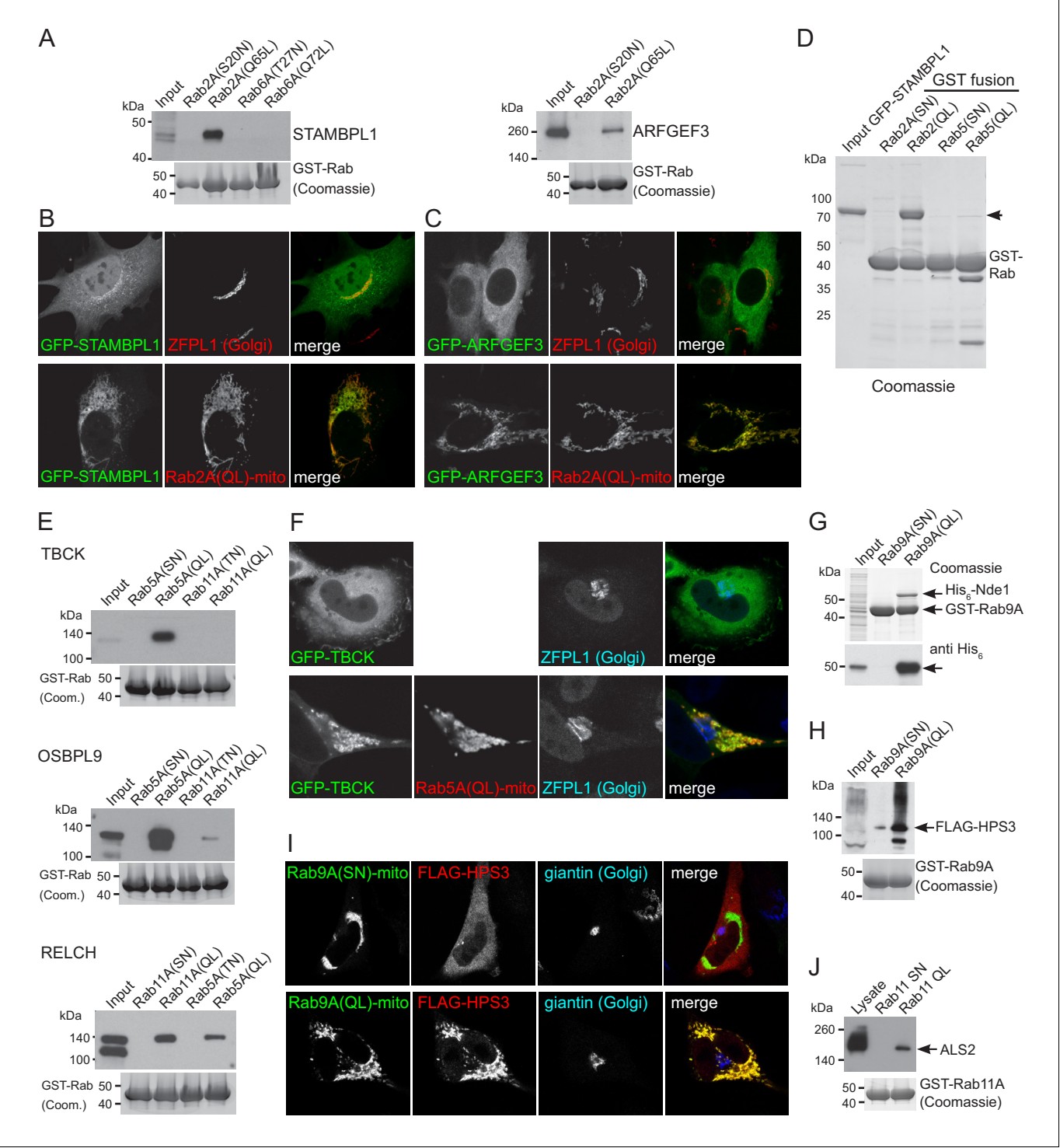

**Figure 7.** Validation of putative effectors for Rab2, Rab5A and Rab9A. (**A**) Affinity chromatography of cell lysates (input) using GST-fusions to the indicated GTP-locked (QL) or GDP-locked Rabs. The bound proteins were eluted and probed with antibodies to the indicated proteins. Lysates were from HEK293 cells and probed for STAMBPL1, or from rat brain and probed for ARFGEF3. (**B**) Confocal micrographs of MEFs expressing either GFP-STAMBPL1 alone and probed for a Golgi marker, or co-expressing the mitochondrial Rab2A(Q65L)-BirA* as indicated. (**C**) As (**B**) except that the cells were expressing GFP-ARFGEF3. (**D**) Coomassie blue stained gel showing affinity chromatography of purified recombinant GFP-STAMBPL1 using GST fusions to the indicate GTP-locked (QL) or GDP-locked (SN) Rabs. The recombinant GFP-STAMBPL1 binds specifically to the GTP form of Rab2A. (**E**) Affinity chromatography of HeLa cell lysates (input) using GST-fusions to the indicated GTP-locked (QL) or GDP-locked Rabs. The bound proteins were eluted and probed with antibodies to the indicated proteins. TBCK, OSBPL9 and RELCH show specific binding to the GTP form of Rab5, with RELCH

*Figure 7 continued on next page*

*Figure 7 continued*
also binding to Rab11-GTP as previously reported (*Sobajima et al., 2018*). (F) Confocal micrographs of cells expressing GFP-TBCK alone, or with co-expression of mitochondrial Rab5A(Q79L)-BirA* as indicated. (G) Coomassie blue stained gel showing affinity chromatography of lysate from *E. coli* expressing His$_6$-Nde1 (input) using GST fusions to GTP-locked (Q66L) or GDP-locked (S21N) Rab9A. The samples were also immunoblotted for the His$_6$ tag. The recombinant His$_6$-Nde1 binds specifically to the GTP form of Rab9A. (H) Affinity chromatography of lysate from HEK293 cells expressing FLAG-HPS3 (Input) using GST-fusions to GTP-locked (QL) or GDP-locked (SN) Rab9A. The bound proteins were eluted and probed for the FLAG tag. (I) Confocal micrographs of HeLa cells expressing FLAG-HPS3 and mitochondrial Rab9A(Q66L)-BirA* (GTP-locked) or Rab9A(S21N)-BirA* (GDP-locked). The GTP-form specifically recruits HPS3 to mitochondria. (J) Affinity chromatography of HEK293 cell lysate (Input) using GTP-locked (QL) or GDP-locked (SN) GST-Rab11A. Bound proteins were eluted and probed for ALS2/alsin.
DOI: https://doi.org/10.7554/eLife.45916.012

## RalB

The Ral GTPases regulate various events at the cell surface including exocytosis and cell migration (*Gentry et al., 2014*). The top ten most significant hits with the GTP form of RalB include six subunits of the exocyst vesicle tethering complex, a known RalB effector, including the EXOC2/Sec5 subunit to which RalB binds directly (*Sugihara et al., 2002*) (*Figure 8C*). The hits also include the Ral effector RLIP/RalBP1 and its binding partner REPS1/Reps1, and the subunits of the two known Ral GAPs (RALGAPA1/2 and RALGAPB) (*Yamaguchi et al., 1997*; *Shirakawa et al., 2009*). The other more significant hits are nucleic acid binding proteins that seem likely to be contaminants. The GDP-bound form of RalB did not give a significant score with any known or plausible GEF.

## Rheb

The small GTPase Rheb is a major regulator of the mTORC1 kinase complex that controls protein synthesis and cell growth (*Heard et al., 2014*). The top hit with the GTP-bound form of Rheb is the mTORC1 subunit Raptor that is located next to the Rheb binding site on mTor (*Yang et al., 2017*) (*Figure 8D*). The next two hits are Tsc1 and Tsc2, the subunits of the Rheb GAP that controls the activity of Rheb and hence mTORC1 (*Manning and Cantley, 2003*). The remaining hits have substantially lower scores. The mechanism by which Rheb is activated is not known, and the hits with the GDP-bound form seem unlikely to be meaningful as they have known functions unrelated to growth control.

## Application of MitoID to Rho GTPases

The Rho family is another of the five sub-families that comprise the Ras superfamily. The Rho GTPases are major regulators of the actin cytoskeleton (*Heasman and Ridley, 2008*; *Narumiya and Thumkeo, 2018*), and we tested the MitoID approach with three of the best characterized members of the family: Cdc42, RhoA, and Rac1. As with the other GTPase we used mutant forms predicted to be locked in the GTP-bound or GDP-bound states, and the findings are summarized below.

## Cdc42

The top 25 most significant hits with the GTP-form of Cdc42 include 19 known Cdc42 effectors (including kinases, actin regulators and scaffolding proteins), two known Cdc42 GAPs (ARHGAP31/32), and myosin18A that has been reported to form a complex with CDC42BPB, one of the Cdc42 effectors (*Tan et al., 2008*; *Vigil et al., 2010*) (*Figure 9A*). The remaining proteins include KCTD3, a putative linker to cullin E3 ligase (*Pinkas et al., 2017*). Its function is unknown, but its paralog SHKBP1 is also present in the list of hits. In addition some hits are regulators of other members of the Rho family such as the RhoA GEFs ARHGEF11/PDX-RhoGEF, ARHGEF12/LARG, ARHGEF40/Solo and PLEKHG4 reflecting the known cross talk between the Rho family GTPases (*Paul et al., 2017*).

With the GDP-locked form of Cdc42 the top hits include four members of the DOCK family of Rho GEFs that are known to activate Cdc42, along with LRCH2 and LRCH3 that have recently been found to interact with a subset of DOCK family GEFs (*O'Loughlin et al., 2018*) (*Figure 9B*). The other hits include ARHGAP32/p250GAP, a Rho family GAP reported to act on Cdc42 and RhoA (*Nakazawa et al., 2003*). ARHGAP32 was found with both GDP and GTP forms of Cdc42 suggesting its substrate recognition is not sensitive to nucleotide state. Interestingly, one of the lower

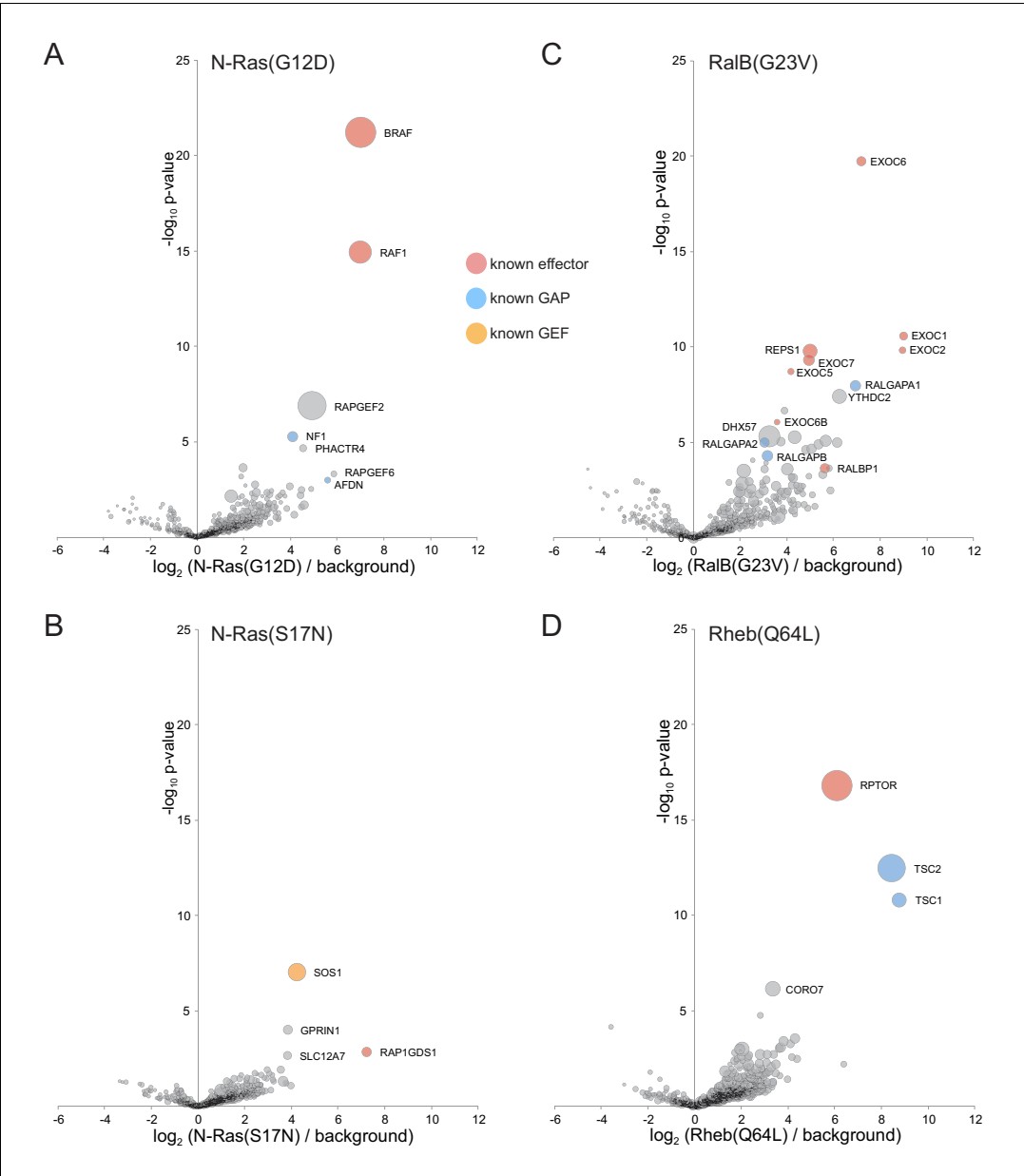

**Figure 8.** Application of MitoID to N-Ras, RalB and Rheb. (**A–F**). Bubble-volcano plots of MitoID with the indicated Ras family GTPases, each with mutations predicted to lock it into the GTP-bound form, with N-Ras also shown with the GDP-bound/empty form. In each case the GTPase is compared to a background comprising the GDP-locked/empty forms of seventeen GTPases, with the size of the bubbles proportional to the WD scores. Indicated are known effectors (red), GAPs (blue) and GEFS (yellow). All values are in *Supplementary file 4*.

DOI: https://doi.org/10.7554/eLife.45916.013

significance hits was PDLIM4, a member of the 'enigma' family of LIM/PDZ domain proteins. These proteins interact with the actin cytoskeleton, with one recently reported to be required for activation of Cdc42 (*te Velthuis and Bagowski, 2007*; *Liu et al., 2015*).

## RhoA

The top 35 hits with GTP-locked RhoA include 19 known effectors including kinases like PKN1 and CIT/Citron, scaffolds like the Rhotekins RTKN/RTKN2, RhoA GEFs such as ARHGEF1/p115-RhoAGEF and ARHGEF12, and phosphatases such as INPPL1/SHIP (*Figure 9C*). In addition, there are six

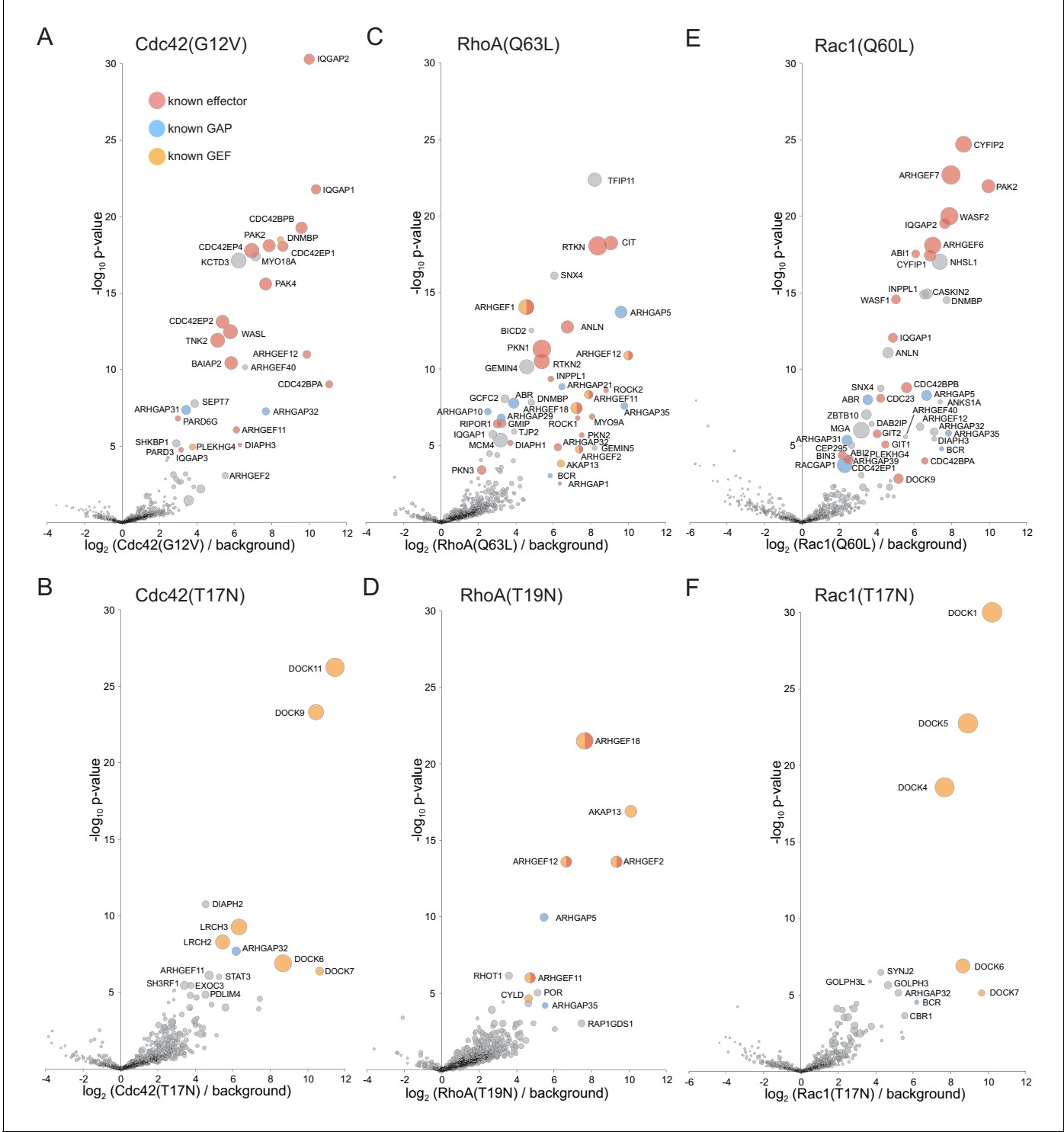

**Figure 9.** Application of MitoID to CDC42, RhoA and Rac1. (**A–F**). Bubble-volcano plots of MitoID with the indicated Rho family GTPase, each with mutations predicted to lock it in to either the GTP-bound (**A,C,E**), or GDP-bound/empty form (**B,D,G**). In each case the GTPase is compared to a background comprising the GDP-locked/empty forms of seventeen GTPases, with the size of the bubbles proportional to the WD scores. Indicated are known effectors (red), GAPs (blue) and GEFS (yellow). All values are in ***Supplementary file 4***.

DOI: https://doi.org/10.7554/eLife.45916.014

The following figure supplement is available for figure 9:

**Figure supplement 1.** MitoID with sequence variants of GTPases.

*Figure 9 continued on next page*

*Figure 9 continued*

DOI: https://doi.org/10.7554/eLife.45916.015

known RhoA GAPs such as ABR, ARHGAP5, ARHGAP35/p190RhoGAP. The remaining strong hits include the dynein adaptor BICD, a sorting nexin, and four DNA or RNA binding proteins (GEMIN4/5, GCFC2 and MCM4) which seem likely to be contaminants. With the GDP-locked form of RhoA the top four hits are all known RhoA GEFs, with further other GEFs with lesser scores (*Figure 9D*). The top four were also hits mentioned above with the GTP-form of RhoA reflecting the fact that they are also effectors that act in a positive feedback loop (*Medina et al., 2013*).

### Rac1

Initial application of MitoID to Rac1 revealed known interactors but also showed Rac1-specific biotinylation of many DNA or RNA binding proteins, with these interactions typically being nucleotide-state independent (*Figure 9—figure supplement 1A*). Some Ras family GTPases have poly-basic stretches next to their C-terminal lipid anchors which stabilize interactions with acidic phospholipids in the cytosolic leaflet of the plasma membrane. On the neutral membranes of mitochondria these basic residues might instead recruit nucleic acids and so we repeated the MitoID with Rac1 lacking the six consecutive basic residues adjacent to the lipid anchor. This reduced the number of nucleic-acid binding proteins but did not prevent interactions with effectors and so our analysis was performed with this version of Rac1 (*Figure 9—figure supplement 1B*).

The top eight hits with Rac1-GTP are all known Rac1 effectors, the GIT/PIX (ARFGEF6/7) complexes that are also Cdc42/Rac1 GEFs, the PAK2 kinase and the WASF2, ABI1 and CYFIP subunits of the WAVE2 actin regulatory complex (*Figure 9E*). The next 30 most significant hits include ten further Rac1 effectors, and six Rac1 GAPs. The remaining proteins include several uncharacterized members of the ARHGEF and ARHGAP families of Rho GTPase regulators, and NHSL1, one of three paralogs of the Nance-Horan syndrome protein. The function of the NHS family of proteins is unknown, but interestingly they have a WAVE homology domain and their knockdown perturbs the actin cytoskeleton (*Brooks et al., 2010*).

With the GDP-locked form of Rac1 the top four hits are members of the DOCK family of Rho GEFs (*Figure 9F*). Interestingly, DOCK6 and DOCK7 were found with both Rac1 and Cdc42 whereas DOCKs 1,4 and 5 were specific to Rac1 and DOCKs 9 and 11 were specific to Cdc42, accurately reflecting their known substrate specificity (*Gadea and Blangy, 2014*).

Taken together these results provide overwhelming evidence that the MitoID approach works not only with Rab GTPases but also with the members of the Ras and Rho families of GTPases.

## Discussion

Proximity biotinylation has emerged as a powerful method for identifying the interaction partners of proteins (*Lambert et al., 2015*; *Roux et al., 2012*). However, like other methods aimed at this goal it generates not just signal but also noise, which in this case comes from bystander proteins that are not direct interactors. Again, as for other methods, specific interactions can be identified by comparing many baits, but this requires the noise to be as uniform as possible. We show here that relocating a set of baits to the same ectopic location can still be compatible with binding to known interactors whilst allowing the advantage of meaningful comparison between baits.

The method has several further advantages over more conventional approaches based on affinity chromatography or yeast two-hybrid screening. Expressing human proteins in cultured cells allows access to relevant chaperones or post-translational modifications, maximizing the chance that the bait is active. The binding is assayed in living cytosol and hence at native protein and ion concentrations. As the protein is at an ectopic location with a large capacity to handle exogenous proteins then over-expression should increase the biotinylation of targets without the risk of accumulation in an undesired location. Finally, it should be noted that the method does not require a very large number of cells. For each replicate we used two T175 flasks totalling $5 \times 10^7$ cells that were transiently transfected with the expression plasmid, and so MitoID should be readily applicable to comparing different cell types.

We have previously used affinity chromatography of cell lysates to screen a panel of *Drosophila* Rabs (*Gillingham et al., 2014*). To compare MitoID with affinity chromatography we determined how many validated effectors for the well characterized Rabs, Rab2 and Rab5, were identified as hits by MitoID versus how many had *Drosophila* orthologs that were found by affinity purification. For Rab5 there are 16 reported mammalian effector proteins or complexes that are conserved in *Drosophila*, of which 14 were found by MitoID and 13 by affinity chromatography of S2 lysate (*Supplementary file 5*). MitoID also found all three of the reported effectors that are not conserved in flies. In addition, several of the hits were not previously reported as effectors, and of these some were also found by affinity chromatography, and we have validated two that were hits with both approaches (OSBPL9 and TBCK) and one present only in mammals (RELCH). For Rab2 there are 12 reported mammalian effector proteins or complexes that are conserved in *Drosophila*, of which 11 were found by MitoID and eight by affinity chromatography of S2 cell lysate (*Supplementary file 5*). As with Rab5 there were several proteins not previously reported to be mammalian effectors that were found by both methods, or only by MitoID, and we validated two of the latter (ARFGEF3 and STAMBPL1). Taken together, this analysis indicates that the MitoID can perform as well, if not better, than affinity chromatography, and illustrates MitoID's ability to find novel effectors for even well characterized Rabs.

However, it is also important to acknowledge the MitoID method has some limitations. Firstly, it is possible that some GTPase effectors will require a specific lipid for stable membrane association, although we were able to find binding partners for GTPases from several different locations that are known to differ in lipid composition. Secondly, it may be harder to detect GTPase effectors that are normally present on mitochondria. The Rab5 effector TBC1D15 and the Rab7 effector Vps13C are known to be associated with mitochondria, and indeed were biotinylated by most of the baits. However, they were also detected on the bubble-volcano plots as putative hits with Rab5 and Rab7 respectively, albeit with low WD scores, indicating that at least in these two cases the extra degree of biotinylation allowed them to emerge from the mitochondrial background. Thirdly, we only found known GEFs with a subset of the Rab GTPases. This may reflect the fact that the canonical 'GDP-locked' mutation used in this study has a complex effect in that it destabilizes the binding of both GTP and GDP but the latter interaction is less affected in the limited number of cases examined (*Koch et al., 2016*; *John et al., 1993*). Although all Ras superfamily members share the same general fold it is known that the precise details of their structure and activation cycle are somewhat variable (*Langemeyer et al., 2014*; *Merithew et al., 2001*; *Cherfils and Zeghouf, 2013*; *Eathiraj et al., 2005*). It has been reported that the GDP-locked form of Rab27A is rapidly degraded in vivo suggesting that the mutation destabilizes the protein, and here we found that GDP forms of Rab7A and Rab18 were expressed at lower levels and did not detect their GEFs (*Ramalho et al., 2002*). It may thus be that successful application of MitoID with the GDP-locked form requires that the empty state is not only stable but also able to bind the GEF sufficiently tightly to allow detection, and it is possible that screening further mutations in the nucleotide binding pocket could facilitate this. Fourthly, although we found known effectors with all of the non-Rab GTPases examined, we did not find high confidence hits with a few Rabs that we tested including Rab7L1, Rab19, and Rab39 and so they were not included in the analysis. These Rabs are known to be much less abundant in HeLa cells than the Rabs presented here and so it is possible that their effectors are too scarce to be detected with the conditions used (*Hein et al., 2015*).

Despite these caveats, the MitoID approach has clear potential for analysing the roles of the many less well characterized members of the Ras superfamily. In addition, there is scope for addressing the few limitations of the method. Our approach relied on using the original BirA* biotin ligase but more active variants have been recently reported (*Branon et al., 2018*; *Kim et al., 2016*). Likewise, the sensitivity of mass-spectrometers is still increasing. Finally, exploring more mutations in the GTPases may identify GTP-locked or GDP-bound/empty forms that act more effectively to capture effectors or GEFs respectively. Indeed, MitoID allows rapid testing of sequence variants - following transient transfection of variants, the biotinylated proteins can be isolated with streptavidin and probed by immunoblotting for known interactors. To illustrate this we tested the effect on Rab5A-GTP of mutations in Ser84 that has been identified as a potential site for phosphorylation (*Steger et al., 2017*; *Ong et al., 2014*). Mutating this to alanine or glutamate greatly reduced biotinylation of one effector (OSBPL9), but not others, suggesting that modification of this residue could differentially affect effector binding (*Figure 9—figure supplement 1C*).

We anticipate that the MitoID approach, and the data presented in this paper, will be of help to many labs investigating small GTPases and potentially other families of proteins that function through transient interactions with cytosolic proteins.

# Materials and methods

## Key resources table

| Reagent type (species) or resource | Designation | Source or reference | Identifiers | Additional information |
|---|---|---|---|---|
| Strain, strain background (*E. coli*) | BL21 GOLD (DE3) | Agilent Technologies | 230132 | Used for expression |
| Strain, strain background (*E. coli*) | Alpha-select gold | Bio-line | BIO-85027 | Used for cloning |
| Cell line (*H. sapiens*) | mCherry-Parkin, HeLa | (*Lazarou et al., 2015*) | | |
| Cell line (*H. sapiens*) | HEK293 | ATCC | ATCC CRL-11268 RRID:CVCL_1926 | |
| Cell line (*H. sapiens*) | HeLa | ATCC | ATCC CCL-2 RRID:CVCL_0030 | |
| Recombinant DNA reagent | pcDNA3.1 | Clontech | V79020 | |
| Recombinant DNA reagent | pGEX6p2 | GE Healthcare Life Sciences | GE28-9546-50 | |
| Detection reagent | Streptavidin-HRP | Cell Signaling Technology | 3999S RRID:AB_10830897 | WB: 1:300 |
| Antibody | Rat monoclonal anti-HA (3F10) | Roche | 867 423 001 RRID:AB_2314622 | IHC: 1:300 |
| Antibody | Goat polyclonal anti-giantin | Santa Cruz | sc-46993 RRID:AB_2279271 | IHC: 1:200 |
| Antibody | Rabbit polyclonal human COXIV (3E11) | New England Biolabs | 4850S RRID:AB_2085424 | IHC: 1:200 |
| Antibody | Rabbit polyclonal anti- ZFPL1 | Sigma | HPA014909 RRID:AB_1859055 | IHC: 1:300 |
| Antibody | Mouse monoclonal anti-TOM20 | BD Transduction Labs | 612278 RRID:AB_399595 | IHC: 1:200 |
| Antibody | Sheep polyclonal anti-TGN46 | ABD serotec | AHP500G RRID:AB_323104 | IHC: 1:300 |
| Antibody | Rabbit polyclonal anti-golgin-84 | Sigma | HPA000992 RRID:AB_1079009 | IHC: 1:300 |
| Antibody | Mouse monoclonal anti-HIS$_6$ tag | Abcam | Ab18184 RRID:AB_444306 | WB: 1:1000 |
| Antibody | Rabbit polyclonal Anti-STAMBPL1 | Sigma | SAB4200146 RRID:AB_10622601 | WB: 1:1000 |
| Antibody | Rabbit polyclonal anti-NDE1 | Life Technologies | 711424 RRID:AB_2692258 | WB: 1:500 |
| Antibody | Rabbit polyclonal anti-TBCK | Cambridge Bioscience | HPA039951 RRID:AB_10795300 | WB: 1:500 |
| Antibody | Rabbit polyclonal anti-RELCH/KIAA1468 | Cambridge Bioscience | HPA040038 RRID:AB_10793860 | WB: 1:1000 |
| Antibody | Rabbit monoclonal anti-OSBPL9 | Abcam | ab151691 | WB: 1:1000 |
| Antibody | Mouse monoclonal anti-FLAG (M2) | Sigma | F1804 RRID:AB_262044 | WB: 1:5000 |

*Continued on next page*

*Continued*

| Reagent type (species) or resource | Designation | Source or reference | Identifiers | Additional information |
|---|---|---|---|---|
| Antibody | Rabbit polyclonal anti-ARFGEF3 | ThermoFisher Scientific | PA5-57623 RRID:AB_2638119 | WB: 1:1000 |
| Antibody | Alexa 488 Donkey polyclonal anti-Rat IgG | ThermoFisher Scientific | A21208 RRID:AB_2535794 | IHC: 1:400 |
| Antibody | Alexa 647 Donkey polyclonal anti-Goat IgG | ThermoFisher Scientific | A32849 RRID:AB_2762840 | IHC: 1:400 |
| Antibody | Alexa 555 Donkey polyclonal anti-Rabbit IgG | ThermoFisher Scientific | A32794 RRID:AB_2762834 | IHC: 1:400 |
| Antibody | Alexa 647 Donkey polyclonal anti-Rabbit IgG | ThermoFisher Scientific | A32795 RRID:AB_2762835 | IHC: 1:400 |
| Antibody | Alexa 555Donkey polyclonal anti-mouse IgG | ThermoFisher Scientific | A32773 RRID:AB_2762848 | IHC: 1:400 |
| Recombinant DNA reagent | STAMBPL1 | Addgene | #22559 | |
| Recombinant DNA reagent | TBCK | Insight Biotechnology | RC203605 | |
| Recombinant DNA reagent | HPS3 | Insight Biotechnology | RC204569 | |
| Recombinant DNA reagent | NDE1 | Stratech Scientific | HG15586 | |
| Recombinant DNA reagent | Rab2AQ65L-BirA*-HA-MAO | This study | pN40 | Deposited at Addgene |
| Recombinant DNA reagent | Rab2AS20N-BirA*-HA-MAO | This study | JB2 | Deposited at Addgene |
| Recombinant DNA reagent | Rab5AQ79L-BirA*-HA-MAO | This study | JB28 | Deposited at Addgene |
| Recombinant DNA reagent | Rab5AS34N-BirA*-HA-MAO | This study | JB40 | Deposited at Addgene |
| Recombinant DNA reagent | Rab6AQ72L-BirA*-HA-MAO | This study | pO40 | Deposited at Addgene |
| Recombinant DNA reagent | Rab6AT27N-BirA*-HA-MAO | This study | JB4 | Deposited at Addgene |
| Recombinant DNA reagent | Rab7AQ67L-BirA*-HA-MAO | This study | JB93 | Deposited at Addgene |
| Recombinant DNA reagent | Rab7AT22N-BirA*-HA-MAO | This study | JB94 | Deposited at Addgene |
| Recombinant DNA reagent | Rab8AQ67L-BirA*-HA-MAO | This study | JB24 | Deposited at Addgene |
| Recombinant DNA reagent | Rab8AT22N-BirA*-HA-MAO | This study | JB23 | Deposited at Addgene |
| Recombinant DNA reagent | Rab9AQ66L-BirA*-HA-MAO | This study | JB84 | Deposited at Addgene |
| Recombinant DNA reagent | Rab9AS21N-BirA*-HA-MAO | This study | JB85 | Deposited at Addgene |
| Recombinant DNA reagent | Rab10Q68L-BirA*-HA-MAO | This study | JB81 | Deposited at Addgene |
| Recombinant DNA reagent | Rab10T23N-BirA*-HA-MAO | This study | JB82 | Deposited at Addgene |

*Continued on next page*

*Continued*

| Reagent type (species) or resource | Designation | Source or reference | Identifiers | Additional information |
|---|---|---|---|---|
| Recombinant DNA reagent | Rab11AQ69L-BirA*-HA-MAO | This study | JB20 | Deposited at Addgene |
| Recombinant DNA reagent | Rab11AS25N-BirA*-HA-MAO | This study | JB19 | Deposited at Addgene |
| Recombinant DNA reagent | Rab11BQ70L-BirA*-HA-MAO | This study | pX49 | Deposited at Addgene |
| Recombinant DNA reagent | Rab11BS25N-BirA*-HA-MAO | This study | pY49 | Deposited at Addgene |
| Recombinant DNA reagent | Rab18Q67L-BirA*-HA-MAO | This study | JB11 | Deposited at Addgene |
| Recombinant DNA reagent | Rab18S22N-BirA*-HA-MAO | This study | JB10 | Deposited at Addgene |
| Recombinant DNA reagent | Rab30Q68L-BirA*-HA-MAO | This study | JB15 | Deposited at Addgene |
| Recombinant DNA reagent | Rab30T23N-BirA*-HA-MAO | This study | JB3 | Deposited at Addgene |
| Recombinant DNA reagent | Rab33BQ92L-BirA*-HA-MAO | This study | JB26 | Deposited at Addgene |
| Recombinant DNA reagent | Rab33BT47N-BirA*-HA-MAO | This study | JB25 | Deposited at Addgene |
| Recombinant DNA reagent | nRasG12D-BirA*-HA-MAO | This study | JB95 | Deposited at Addgene |
| Recombinant DNA reagent | nRasS17N-BirA*-HA-MAO | This study | JB96 | Deposited at Addgene |
| Recombinant DNA reagent | Rac1Q60L-BirA*-HA-MAO | This study | JB42 | Deposited at Addgene |
| Recombinant DNA reagent | Rac1T17N-BirA*-HA-MAO | This study | JB54 | Deposited at Addgene |
| Recombinant DNA reagent | RalBG23V-BirA*-HA-MAO | This study | JB97 | Deposited at Addgene |
| Recombinant DNA reagent | RalBS29A-BirA*-HA-MAO | This study | JB98 | Deposited at Addgene |
| Recombinant DNA reagent | RhoAQ63L-BirA*-HA-MAO | This study | JB68 | Deposited at Addgene |
| Recombinant DNA reagent | RhoAT19N-BirA*-HA-MAO | This study | JB55 | Deposited at Addgene |
| Recombinant DNA reagent | Cdc42G12V-BirA*-HA-MAO | This study | JB48 | Deposited at Addgene |
| Recombinant DNA reagent | Cdc42T17N-BirA*-HA-MAO | This study | JB57 | Deposited at Addgene |
| Recombinant DNA reagent | RhebQ64L-BirA*-HA-MAO | This study | JB87 | Deposited at Addgene |
| Recombinant DNA reagent | RhebS20N-BirA*-HA-MAO | This study | JB88 | Deposited at Addgene |
| Peptide, recombinant protein | 3X FLAG peptide | Sigma | F4799 | |
| Commercial assay or kit | Amersham ECL Prime detection agent | GE Healthcare Lifesciences | RPN2232 | Western blots |
| Commercial assay or kit | Supersignal West Femto Maximum Sensitivity Substrate | Thermo Scientific | 34095 | Western blots |

*Continued on next page*

*Continued*

| Reagent type (species) or resource | Designation | Source or reference | Identifiers | Additional information |
|---|---|---|---|---|
| Chemical compound | Biotin | Sigma | B4501 | |
| Software, algorithm | Perseus | (*Tyanova et al., 2016*). | RRID:SCR_015753 | |
| Other | Dynabeads MyOne Streptavidin C1 | Invitrogen | 65002 | Isolation of biotinylated proteins |
| Other | Anti-FLAG M2 affinity resin | Sigma | A2220 RRID:AB_10063035 | Anti-FLAG immunoprecipitation |
| Other | Fugene 6 | Promega | E2691 | Transfection |
| Other | Neutravidin-FITC | Invitrogen | 31006 | |
| Other | Glutathione sepharose 4B | GE Healthcare Lifesciences | 17075601 | GST affinity chromatography |
| Other | Vectashield | Vector Laboratories Inc | H-1000 | Mounting media |
| Other | Novex 4–20% Tris-Glycine gels | Thermofisher | XP04202BOX | Pre-cast gels |
| Other | Gibco Opti-MEM | Fisher Scientific | 31985070 | Media |

## Plasmids

For MitoID, Ras family GTPases lacking C-terminal cysteine residues and with mutations locking them in either the active or inactive conformation were cloned into pcDNA3.1$^+$ (Clontech), upstream of a GAGA linker. This was followed by the BirA* ligase, a second GAGAGA linker, a HA tag and residues 481–527 of monoamine oxidase (MAO) to target the construct to the mitochondria. The MitoID plasmids reported here are available from Addgene (https://www.addgene.org/Sean_Munro/).

Human STAMBPL1 was PCR amplified from FLAG-HA-STAMBPL1 (Addgene plasmid #22559), human TBC1 domain containing kinase (TBCK) was PCR amplified from Myc-DDK-tagged TBCK (transcript variant 4) (Insight Biotechnology Ltd, RC203605) and human Hermansky-Pudlak syndrome 3 (HPS3) from Myc-DDK-tagged HPS3 (Insight Biotechnology Ltd, RC204569). All were cloned between Not1 and XmaI downstream of FLAG or GFP-FLAG tags in pcDNA3.1$^+$. GFP-FLAG- constructs have a GAGAGA linker between the GFP and the FLAG tag. FLAG tag constructs have a GAGAGA linker and additional Prescission protease site the tag and the insert. ARFGEF3 was custom synthesized (Epoch Life Sciences) and cloned between KpnI and NotI into pcDNA3.1$^+$. FLAG or GFP-FLAG tags as above were PCR amplified and inserted upstream of the ARFGEF3 ORF between NheI and KpnI. Bacterial expression used *E.coli* codon-optimized human Rab constructs created by cloning Rab GTPases lacking the C-terminal cysteines and with the relevant activating or inactivating point mutations (see Key resources table) into the vector pGEX6p2 between EcoR1 and XhoI (GE Healthcare Life Sciences). Full-length NDE1 was PCR amplified from the plasmid HG15586 (Stratech Scientific) and cloned between BamH1 and HindIII in *E. coli* expression vector p810 (Andrew Carter, MRC LMB Cambridge UK), thus adding an N-terminal His$_6$ tag followed by a dihydrolipoyl acetyl-transferases solubility tag and a TEV cleavage site. The GFP-PX domain plasmid was provided by Yohei Ohashi (MRC LMB, Cambridge, UK).

## Antibodies and probes

Antibodies used for immunofluorescence experiments in this study were: HA tag (3F10/11 Roche, 867 423 001), giantin (Santa Cruz, N-18/sc-46993), COXIV (3E11, New England Biolabs Ltd, 4850S), CI-MPR (Abcam ab124767), golgin-84 (Sigma HPA000992), TOM20 (BD Transduction Labs, 612278) and ZFPL1 (Sigma, HPA014909). Endosomes were labeled with Alexa Fluor 555 transferrin (Thermo-Fisher Scientific, T35352). Antibodies for Western blotting were the same as those used for immunofluorescence plus: STAMBPL1 (Sigma, SAB4200146), TBCK (Cambridge Bioscience, HPA039951), RELCH/KIAA1468 (Cambridge Bioscience, HPA040038), OSBPL9 (Abcam, ab151691), PIK3R4 (Abcam, ab128903), FLAG (M2) (Sigma, F1804), EEA1 (Abcam, ab109110), ARFGEF3 (ThermoFisher

Scientific, PA5-57623), ALS2 (Abcam, ab170896) and α-tubulin (YL1/2; *Kilmartin et al., 1982*). Biotin was detected with Neutravidin-FITC (Invitrogen, 31006) and Streptavidin-HRP (Cell Signaling Technology, 3999S).

## Cell culture and immunofluorescence

All cells were cultured in Dulbecco's modified Eagle's medium (DMEM; Invitrogen) supplemented with 10% fetal calf serum (FCS) and penicillin/streptomycin at 37°C and 5% CO2. All cell lines were regularly tested to confirm that they were free from mycoplasma using the MycoAlert kit (Lonza). HeLa cells expressing mCherry-parkin were described previously (*Lazarou et al., 2015*). For immunofluorescence experiments, HeLa cells (ATCC) were transfected with 1 µg of DNA and 4 µL FuGENE 6 in 100 µL Opti-MEM media for 24–36 hr according to the manufacturer's instructions (Promega). Cells were fixed with 4% (v/v) formaldehyde in PBS and permeabilized in 0.5% (v/v) Triton-X-100 in PBS. Cells were blocked for one hour in PBS containing 20% (v/v) FCS and 0.25% (v/v) Tween-20 and probed with the antibodies in the same buffer. Primary antibodies were detected with species-specific Alexa Fluor-labeled secondary antibodies (Molecular Probes). The cells were mounted in Vectashield (Vector Laboratories) and imaged using either an LSM 780 (Zeiss) or a TCS SP8 (Leica) confocal microscope.

## Protein blotting

All blots, except for those with Streptavidin-HRP, were blocked in 5% (w/v) milk in PBS-T (PBS with 0.1% (v/v) Tween-20) for 30–60 min, incubated either for 1 hr at room temperature or overnight at 4°C with primary antibody in the same blocking solution. Washed extensively with PBS-T then incubated with HRP-conjugated secondary antibody in 0.1% (w/v) milk in PBS-T for one hour, washed again in PBS-T, then once in PBS only and detected with either Immobilon Western HRP substrate or Amersham ECL detection reagent. For probing with Streptavidin-HRP, the blots were blocked in 0.2% (w/v) I-BLOCK (Tropix, Applied Biosystems) in 0.1% PBS-T for 30–60 min. Primary and secondary antibodies were diluted using this blocking solution and incubated and processed as above.

## Affinity capture of biotinylated proteins

The protocol for isolating proteins biotinylated by BirA* was adapted from the BioID method (*Roux et al., 2012*). Briefly, HEK293T cells were grown in two 175 cm$^2$ flasks to ~50% confluence and transfected with 25 µg plasmid and 75 µl FuGENE 6 in 2 ml Opti-MEM according to the manufacturer's instructions (Promega). One day after transfection, biotin was added to 50 µM, and the cells incubated for a further 18 hr. Cells were pelleted by centrifugation (1000 x *g*, 5 min), washed once in ice-cold PBS and resuspended in lysis buffer (25 mM Tris pH 7.4, 150 mM NaCl, 1 mM EDTA, 1% (v/v) Triton X-100, 1 mM PMSF, 1 cOmplete protease inhibitor cocktail tablet (Roche)/50 ml buffer), and incubated for 30–60 min at 4°C with rotation. After centrifugation at 10,000 x *g* for 10 min at 4°C, the supernatants were added to 500 µl Dynabeads MyOne Streptavidin C1 beads (Invitrogen) that had been pre-washed twice in the same buffer. The beads were incubated at 4°C overnight, washed twice in Wash Buffer 1 (2% SDS PAGE, cOmplete inhibitors), three times in Wash Buffer 2 (1% (v/v) Triton X-100, 0.1% (w/v) deoxycholate, 500 mM NaCl, 1 mM EDTA, 50 mM HEPES, cOmplete inhibitors, pH 7.5), and three times in Wash Buffer 3 (50 mM Tris pH 7.4, 50 mM NaCl, cOmplete inhibitors). Finally, the beads were incubated in 75 µl SDS sample buffer containing 3 mM biotin at 98°C for 5 min to release the biotinylated proteins from the beads. 1 mM β-mercaptoethanol was then added to the SDS sample buffer and 20 µl of the sample analysed by SDS-PAGE and mass spectrometry with the remainder reserved for immunoblotting. All MitoID experiments were performed as biological replicates: the three experiments that constitute a triplicate set for a given GTPase performed on cells transfected independently on different days and processed separately.

## Mass spectrometry

Samples obtained from affinity chromatography and proximity biotinylation were loaded on 4–20% Tris-glycine SDS-PAGE gels and run for 1–2 centimeters. Proteins were stained with Coomassie InstantBlue (Expedeon), the protein-containing part of the gel lane cut into eight slices that were placed in a 96-well plate and destained with 50% v/v acetonitrile and 50 mM ammonium bicarbonate, reduced with 10 mM DTT, and alkylated with 55 mM iodoacetamide. Digestion was with 6 ng/µl

trypsin (Promega, UK) overnight at 37°C, and peptides extracted in 2% v/v formic acid 2% v/v aceto-nitrile, and analyzed by nano-scale capillary LC-MS/MS (Ultimate U3000 HPLC, Thermo Scientific Dionex) at a flow of ~300 nL/min. A C18 Acclaim PepMap100 5 µm, 100 µm x 20 mm nanoViper (Thermo Scientific Dionex), trapped the peptides prior to separation on a C18 Acclaim PepMap100 3 µm, 75 µm x 250 mm nanoViper. Peptides were eluted with an acetonitrile gradient. The analytical column outlet was interfaced via a nanoflow electrospray ionization source with a linear ion trap mass spectrometer (Orbitrap Velos, Thermo Scientific). Data dependent analysis was performed using a resolution of 30,000 for the full MS spectrum, followed by ten MS/MS spectra in the linear ion trap. MS spectra were collected over a m/z range of 300–2000. MS/MS scans were collected using a threshold energy of 35 for collision-induced dissociation. The mass spectrometry proteomics data have been deposited to the ProteomeXchange Consortium via the PRIDE partner repository with the dataset identifier PXD013668 (*Perez-Riverol et al., 2019*).

## Analysis of mass spectrometry data

For analysis of spectral counts and calculation of WD scores LC-MS/MS data were searched against the manually reviewed UniProt human proteome using Mascot (Matrix Science), with a precursor tolerance of 5 ppm and a fragment ion mass tolerance of 0.8 Da. The gene RABGAP1L, has two entries, RBG1L_HUMAN and RBG10_HUMAN, and so the latter as removed. Two missed enzyme cleavages and variable modifications for oxidized methionine, carbamidomethyl cysteine, pyrogluta-mic acid, phosphorylated serine, threonine and tyrosine were included. MS/MS data were validated using the Scaffold programme (Proteome Software Inc). To score the significance of interactors total spectral counts which were converted into D- and WD- scores according to the CompPASS methods (*Sowa et al., 2009*). The D-score assigns more confidence to proteins that are found in replicate experiments and that interact with fewer baits (in this case, fewer Rabs) and is, thus, a measure of specificity and reproducibility. The WD-score, in addition, takes into account that some preys (effec-tors) may interact with a subset of related baits (Rabs) and so in this case the total spectral counts found in this sub-set of baits will be higher than the general background level.

For analysis of spectral intensities and generation of volcano plots LC-MS/MS raw files were proc-essed in MaxQuant (version1.6.2.6) and the peptide lists generated searched against the reviewed UniProt human proteome (as above) using the Andromeda search engine embedded in MaxQuant (*Cox and Mann, 2008*; *Cox et al., 2011*). Enzyme specificity for trypsin was selected (cleavage at the C-terminal side of lysine and arginine amino acid residues, unless proline is present on the car-boxyl side of the cleavage site) and a maximum of two missed cleavages were allowed. Cysteine car-bamidomethylation was set as a fixed modification, while phosphorylation of serine, threonine and tyrosine, and oxidation of methionine were set as variable modifications. Peptides were identified with an initial precursor mass deviation of up to 10 ppm and a fragment mass deviation of 0.2 Da. For label-free protein quantitation (MaxLFQ) we required a minimum ratio count of 1, with two mini-mum and two average comparisons, which enabled normalization of this large dataset (*Hein et al., 2015*). A false discovery rate (FDR), determined by searching a reverse sequence database, of 0.01 was used at both the protein and peptide level.

Data from the Maxquant analysis was analyzed on the Perseus platform (*Tyanova et al., 2016*). Protein identifications were filtered, removing hits to the reverse decoy database as well as proteins only identified by modified peptides. We required that each protein be detected in at least two out of the three replicates from the AP-MS samples of at least one bait. Protein LFQ intensities were log-arithmized and missing values imputed by values simulating noise around the detection limit using the Perseus default settings (*Tyanova et al., 2016*). To calculate p-values two sample Student's t-tests were performed in which baits were compared against the entire set of GDP-locked GTPases with the number of randomizations set at 250.

## Immunoprecipitation of FLAG-tagged STAMBPL1 from HEK293T cells

Three 175 cm$^2$ flasks of HEK293T cells (ATCC) at 60–70% confluence were transfected with plasmids encoding GFP-FLAG tagged STAMBPL1 using FuGENE six according to the manufacturer's instruc-tions. After 48 h cells were harvested by centrifugation, washed once in ice-cold PBS and lysed in lysis buffer (25 mM Tris-HCl pH7.4m 150 mM NaCl, 1 mM EDTA, 0.5% (V/V) Triton X-100, 1 EDTA-free complete protease inhibitor tablet/50 ml, 1 mM PMSF) for 30 min at 4°C. Lysates were clarified

by centrifugation before GFP-FLAG-tagged proteins were isolated by incubation with 100 µL packed FLAG M2 affinity resin (Sigma) for 1 hr with rotation at 4°C. Beads were washed extensively in lysis buffer and bound proteins eluted in 500 µL 100 µg/ml 3X FLAG peptide (Sigma) in 25 mM Tris-HCl pH 7.4, 250 mM NaCl, 1 mM EDTA.

### Affinity chromatography of GST-Rab2A and Rab5A with purified GFP-FLAG-STAMBPL1

Rab2A with Q65L (GTP-locked) or S20N (GFP-locked) mutations and Rab5A with Q79L (GTP-locked) or S34N (GFP-locked) mutations were expressed in *E. coli* strain BL21-GOLD (DE3; Agilent Technologies) as fusions to GST. Bacteria were grown at 37°C to an $OD_{600}$ of 0.7 and induced with 100 µM IPTG overnight at 16°C. Cells were harvested by centrifugation, dounce homogenized and sonicated in lysis buffer (50 mM Tris-HCl, pH 7.4, 150 mM NaCl, 5 mM $MgCl_2$, 1% Triton X-100, 5 mM β-mercaptoethanol, plus 1 EDTA-free complete protease tablet/50 ml, 1 mM PMSF and either 100 µM non-hydrolyzable GTP analog (GppNHp, Sigma) or 100 µM GDP as appropriate). The lysates were clarified by centrifugation at 12,000 x *g* for 15 min and GST-Rab proteins were applied at saturating levels to glutathione Sepharose beads (GE Healthcare) for 30 min at 4°C and then washed extensively to remove unbound material. 100 µl of purified GFP-FLAG-STAMBPL1 was applied to 50 µl of GST-Rab beads in 500 µl lysis buffer plus 100 µM GppNHp or GDP as required, and rotated at 4°C for 2 hr. Beads were then washed in lysis buffer containing 100 µM of the relevant nucleotide before bound proteins were eluted in high salt buffer (25 mM Tris-HCL, pH7.4, 1.5 M NaCl, 20 mM EDTA, 5 mM β-mercaptoethanol and 1 mM of the opposing nucleotide). Proteins were precipitated with chloroform/methanol, resuspended in SDS sample buffer with 1 mM β-mercaptoethanol and analysed by SDS-PAGE.

### Affinity chromatography of *E. coli* expressed GST-Rab9A and His6-tagged NDE1

Rab9A bearing Q66L or S21N mutations was expressed in the *E. coli* strain BL21-GOLD (DE3; Agilent Technologies) as a fusion to GST. NDE1 was expressed in the same bacteria as a fusion to a N-terminal $His_6$ tag with a dihydrolipoyl acetyltransferase solubility tag and a TEV cleavage site between the $His_6$ tag and the NDE1 insert. Bacteria were grown, harvested, lysed and the GST-Rab9A proteins applied to glutathione-Sepharose beads as described above for GST-Rab2A and GST-Rab5A. Lysates containing NDE1 were also incubated with glutathione-Sepharose beads to pre-clear the sample prior to incubation with 50 µl GST-Rab coated beads for 2 hr with rotation at 4°C. Beads and protein complexes were washed extensively in lysis buffer and proteins were eluted in SDS sample buffer with 1 mM β-mercaptoethanol. Lysates and eluates were separated by SDS-PAGE electrophoresis and analyzed either by mass spectrometry or by transfer to nitrocellulose and probing with antibodies.

### Affinity chromatography of cell lysates using GST-Rab2A, 5A, 6A, 9A and 11A

GST-Rab proteins were expressed and purified from the *E. coli* strain BL21-GOLD (DE3; Agilent Technologies) as described above. HEK293T cells from 5 x T175 cm flasks were collected, washed and lysed in 10 ml of lysis buffer (25 mM Tris-HCl, pH 7.4, 150 mM NaCl, 5 mM $MgCl_2$, 1% Triton X-100, 5 mM β -mercaptoethanol, plus 1 EDTA-free complete protease tablet/50 ml and 1 mM PMSF). The lysate was divided equally and applied to 50 µl of GST-Rab coated beads with either 100 µM non-hydrolyzable GTP (GppNHp) or 100 µM GDP added as appropriate. Beads were incubated, washed and proteins eluted in high salt buffer as described above. Proteins were precipitated and analyzed by immunoblotting with the indicated antibodies. As we found no suitable antibody for HPS3, 1 x T175cm² flask of HEK293T cells was transfected with FLAG-tagged HPS3 using Fugene 6 and 24 hr later cells were lysed and incubated with GST-Rab as for the untransfected lysate. In the case of ARFGEF3, rat brain was harvested rapidly and placed into ice-cold PBS. Following several washes in PBS, the brain was minced into small pieces and 10 volumes (50 ml) of lysis buffer added (20 mM Tris-HCl, pH8, 150 mM KCl, 5 mM $MgCl_2$,1 mM PMSF, 1% (w/v) CHAPS, plus 1 EDTA-free complete protease tablet/25 ml buffer). The material was dounce homogenized, solubilized by rotation at 4°C for 3 hr and centrifuged at 100,000 x *g* for 60 min at 4°C. The supernatant was stored in

aliquots at −80˚C. For each Rab affinity chromatography experiment five ml of supernatant was applied to 50 µl of Rab-saturated glutathione Sepharose beads. The beads were then incubated and processed as already described except 1% (w/v) CHAPS replaced Triton X-100 in the wash buffer.

## Acknowledgements

We thank Alan Gillingham, Harriet Parsons, Mark Skehel and Tim Stevens for help with computational analysis of mass-spectrometry data; Wanda Kukulski for advice on mitochondrial stress, and Andrew Carter and Yohei Ohashi for reagents. This work was supported by the Medical Research Council (MRC file reference number MC_U105178783).

## Additional information

### Funding

| Funder | Grant reference number | Author |
|---|---|---|
| Medical Research Council | MC_U105178783 | Alison K Gillingham<br>Jessie Bertram<br>Farida Begum<br>Sean Munro |

The funders had no role in study design, data collection and interpretation, or the decision to submit the work for publication.

### Author contributions

Alison K Gillingham, Formal analysis, Supervision, Validation, Investigation, Project administration, Writing—review and editing; Jessie Bertram, Validation, Investigation; Farida Begum, Investigation; Sean Munro, Conceptualization, Formal analysis, Supervision, Funding acquisition, Methodology, Project administration, Writing—review and editing

### Author ORCIDs

Sean Munro (iD) https://orcid.org/0000-0001-6160-5773

### Decision letter and Author response

Decision letter https://doi.org/10.7554/eLife.45916.025
Author response https://doi.org/10.7554/eLife.45916.026

## Additional files

### Supplementary files

• Supplementary file 1. Analysis of mass-spectral data from GTPase MitoID based on spectral counts. Proteins identified by MitoID with at least one GTPase. Total spectral counts for biological triplicates of GTP-bound (T) and GDP-bound forms (D) are in Sheet S1A. Spectral counts were converted into D scores and WD scores based on the CompPASS platform. D scores in Sheet S1B, mean D scores in Sheet S1C, WD scores in Sheet S1D and mean WD scores in Sheet S1E (see tabs at bottom of sheet). Excel (.xlsx) file.
DOI: https://doi.org/10.7554/eLife.45916.016

• Supplementary file 2. Analysis of mass-spectral data from GTPase MitoID based on peak intensities. Label free quantitation (LFQ) intensities from MaxQuant are shown for all proteins identified by MitoID with at least one GTPase. Excel (.xlsx) file.
DOI: https://doi.org/10.7554/eLife.45916.017

• Supplementary file 3. Volcano plot data from mass spectral peak intensities from MitoID with GTPases of the Rab family. For each Rab form indicated in the tabs, the LFQ intensities of every protein found in at least two of the triplicates was compared to the values obtained with the GDP forms of all GTPases by using the Perseus platform. The difference, expressed as $\log_2$ of the ratio, and the P-value for the significance of the difference are shown, with these being used to generate the

volcano plots shown in the figures. Also shown are the WD score for each interaction as obtained from analysis of spectral counts. Excel (.xlsx) file.

DOI: https://doi.org/10.7554/eLife.45916.018

• Supplementary file 4. Volcano plot data from mass spectral peak intensities from MitoID with GTPases of the Rho and Ras families. For each GTPase form indicated in the tabs, the LFQ intensities of every protein found in at least two of the triplicates was compared to the values obtained with the GDP forms of all GTPases by using the Perseus platform. The difference, expressed as $\log_2$ of the ratio, and the P-value for the significance of the difference are shown, with these being used to generate the volcano plots shown in the figures. Also shown are the WD score for each interaction as obtained from analysis of spectral counts. Excel (.xlsx) file.

DOI: https://doi.org/10.7554/eLife.45916.019

• Supplementary file 5. Coverage of previously reported effectors of mammalian Rab2A and Rab5A by MitoID and S2 cell affinity chromatography. Previously reported effectors for mammalian Rab2 and Rab5 are listed along with their coverage by MitoID and by a previous screen for Rab effectors based on affinity chromatography of *Drosophila* S2 cell lysates (*Gillingham et al., 2014*). Additional tables show for the MitoID interactions the comparisons of LFQ intensities and the WD scores as in *Supplementary file 4*. This illustrates typical such scores for bona fide effectors. The FAM71 family (also called the GARI family) have also been reported to bind human Rab2 (*Fukuda et al., 2008*). However, this interaction is specific for Rab2B and not the Rab2A used for MitoID, and the family is only present in vertebrates, and so it is not included in the comparison. Excel (.xlsx) file.

DOI: https://doi.org/10.7554/eLife.45916.020

• Transparent reporting form

DOI: https://doi.org/10.7554/eLife.45916.021

## Data availability

The mass spectrometry proteomics data have been deposited to the ProteomeXchange Consortium via the PRIDE partner repository with the dataset identifier PXD013668. Apart from this, all data generated or analysed during this study are included in the manuscript and supporting files.

The following dataset was generated:

| Author(s) | Year | Dataset title | Dataset URL | Database and Identifier |
|---|---|---|---|---|
| Skehel M, Munro S | 2019 | In vivo identification of GTPase interactors by mitochondrial relocalizationand proximity biotinylation | https://www.ebi.ac.uk/pride/archive/projects/PXD013668 | EBI PRIDE, PXD013668 |

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
