## [Decision Letter]

Thank you for submitting your article "in vivo identification of GTPase interactors by mitochondrial relocalization and proximity biotinylation" for consideration by *eLife*. Your article has been reviewed by three peer reviewers, one of whom is a member of our Board of Reviewing Editors, and the evaluation has been overseen Anna Akhmanova as the Senior Editor. The following individuals involved in review of your submission have agreed to reveal their identity: Christian Ungermann (Reviewer #2) and Peter S McPherson (Reviewer #3).

The reviewers have discussed the reviews with one another and the Reviewing Editor has drafted this decision to help you prepare a revised submission.

Summary:

Gillingham et al. develop and optimize a novel strategy to identify effectors and regulators of the small GTPases family by taking advantage of the BioID method. Briefly, they fused small GTPases of interest to a BirA tag and localized them to mitochondria using a mitochondrial targeting sequence. At the mitochondrial surface, any protein that comes in close proximity to the small GTPases are biotinylated and subsequently identified by mass spectrometry. By investigating many small GTPases, they were able to distinguish between bona fide interactions and background. They found an elegant way to present their mass spectrometry data and validated some novel interaction between Rabs and effectors by more conventional methods.

Essential revisions:

All three reviewers agreed that the mito ID approach described here is an important extension of the initial analysis carried out previously by the authors using affinity chromatography. Thus, all reviewers support publication provided the following points are addressed during revision:

1) Coverage of the effector spectrum:

In the Gillingham et al., 2014 study, the authors already identify several effectors. Now, each approach – affinity purification and mitoID – has its merits, yet the authors do not explicitly compare their findings here and before. The reviewers suggest to use selected examples like Rab5 or Rab2 to compare the identified effectors, in particular in cases where many effectors are known, thus allowing to obtain an idea how comprehensive the new mitoID method is. We are aware that this compares now S2 cells (*Drosophila*, 2014 study) with human cells (HEK293), yet several effectors are conserved. A Discussion should also include then a comparative evaluation of each method. Moreover, one of the reviewers was surprised that the Rab2 interaction with Vps39 did not show up in their study, even though it was one of the main findings in Gillingham et al., 2014, and later confirmed as an important interaction in autophagy (see Kajiho et al., 2016: Lorincz et al., 2017; Fujita et al., 2017).

2) Expression levels:

Different Rabs are expressed transiently. This will lead to highly variable expression levels. This could strongly influence the background and range of effectors identified. The authors should provide some measure of expression levels for each construct. Perhaps the limitations with the use of GDP-locked forms result simply from poor expression.

3) Suitability of the method for identifying regulators (GEFs, GAPs)

The method appears to be powerful for identifying effectors but mostly ineffective for regulators (especially GEFs). As outlined in the Introduction, numerous effectors have already been identified through yeast two hybrid and affinity chromatography. However, the field is lacking a method to properly identify GEFs/GAPs. Only a small subset of Rab::GDP could identify a GEF. How many Rab::GDP proteins were actually tested? Does it mean the GDP-locked mutant is not appropriate? Would there be other ways to identify GEFs? It would be interesting to validate the interaction between one of the newly identified Rab2::GDP or Rab11::GDP potential GEFs.

---

## [Author Response]

Essential revisions:All three reviewers agreed that the mito ID approach described here is an important extension of the initial analysis carried out previously by the authors using affinity chromatography. Thus, all reviewers support publication provided the following points are addressed during revision:1) Coverage of the effector spectrum:In the Gillingham et al., 2014 study, the authors already identify several effectors. Now, each approach – affinity purification and mitoID – has its merits, yet the authors do not explicitly compare their findings here and before. The reviewers suggest to use selected examples like Rab5 or Rab2 to compare the identified effectors, in particular in cases where many effectors are known, thus allowing to obtain an idea how comprehensive the new mitoID method is. We are aware that this compares now S2 cells (Drosophila, 2014 study) with human cells (HEK293), yet several effectors are conserved. A Discussion should also include then a comparative evaluation of each method. Moreover, one of the reviewers was surprised that the Rab2 interaction with Vps39 did not show up in their study, even though it was one of the main findings in Gillingham et al., 2014, and later confirmed as an important interaction in autophagy (see Kajiho et al., 2016: Lorincz et al., 2017; Fujita et al., 2017).

a) We agree that a detailed comparison of the MitoID method to our previous work and that of others would be a valuable addition. We have followed the suggestion of basing this on Rab5 or Rab2, by adding analysis for both of these Rabs. In particular we compiled lists of all validated mammalian effectors for the two Rabs, and compared how many were identified as hits by MitoID versus how many had *Drosophila* orthologs that were found by affinity purification from S2 cell lysate.

For Rab5 there are 16 reported mammalian effector proteins or complexes that are conserved in *Drosophila*, and of these 14 were found by MitoID and 13 by affinity chromatography of S2 lysate. MitoID also found all three of the reported effectors that are not conserved in flies. In addition, there were several hits that are not previously reported effectors, some of which were also found in S2 cells, and we have validated two that are in both (OSBPL9 and TBCK) and one that is only present in mammals (RELCH).

For Rab2 there are 12 reported mammalian effector proteins or complexes that are conserved in *Drosophila*, of which 11 were identified by MitoID (the exception being the HOPS complex, as discussed below) and 8 by affinity chromatography of S2 lysate. As for Rab5 there were several proteins not previously reported as mammalian effectors that were found with both methods, or only by MitoID, and we validated two of the latter (ARFGEF3 and STAMBPL1).

Taken together, this analysis indicates that the MitoID does at least as well, if not better, that the S2 affinity chromatography approach, and also illustrates the fact that MitoID can find novel effectors for even well characterised Rabs. In addition, it should be emphasised that having made extensive use of both the MitoID and affinity chromatography methods we can state with the confidence that MitoID is simpler and quicker to perform. We have thus added tables showing full details of these comparisons (Supplementary file 5), and added text to the Discussion section to summarise these comparisons. This a very useful addition to the paper and we are grateful to the reviewers for suggesting it.

b) The reviewers noted that we did not find the HOPS complex as an interaction partner with human Rab2, even though we previously found the complex binding to *Drosophila* Rab2 by S2 cell affinity chromatography with the interaction being subsequently reproduced by other labs in flies and mammals. This is the only reported mammalian Rab2 effector that we did not find with MitoID. However, it is worth noting that of the three papers cited by the reviewers, only one was based on mammalian cells (the other two being based on *Drosophila*). In the mammalian study the authors reported finding binding of Vps39 to GST-Rab2, but they also tested Vps41 and did not find an interaction which is puzzling as both Vps39 and Vps41 should be in the same complex. In our own unpublished work we have also been unable to detect the mammalian HOPS complex binding to GST-Rab2 by affinity chromatography of mammalian cells extracts, even though other effectors can be detected. We have not commented on this is the text, beyond noting the lack of HOPS as a hit, as it is clear that overall MitoID outperformed S2 cell affinity chromatography for Rab2, but we thought that we should state here in case it is of interest to the reviewer.

2) Expression levels:Different Rabs are expressed transiently. This will lead to highly variable expression levels. This could strongly influence the background and range of effectors identified. The authors should provide some measure of expression levels for each construct. Perhaps the limitations with the use of GDP-locked forms result simply from poor expression.

We have added immunoblots showing the expression levels of all of the GTPases assayed in the paper (Figure 1—figure supplement 2). We agree that this is a useful addition as it shows that for most GTPases the expression levels of the GTP and GDP forms are similar. There are three cases where the GDP-locked form is apparently less stable, as has been observed previously for some Rabs, and this is now noted in the text.

3) Suitability of the method for identifying regulators (GEFs, GAPs)The method appears to be powerful for identifying effectors but mostly ineffective for regulators (especially GEFs). As outlined in the Introduction, numerous effectors have already been identified through yeast two hybrid and affinity chromatography. However, the field is lacking a method to properly identify GEFs/GAPs. Only a small subset of Rab::GDP could identify a GEF. How many Rab::GDP proteins were actually tested? Does it mean the GDP-locked mutant is not appropriate? Would there be other ways to identify GEFs? It would be interesting to validate the interaction between one of the newly identified Rab2::GDP or Rab11::GDP potential GEFs.

We are pleased that the reviewers feel that our MitoID method is powerful for identifying effectors, and whilst it is true that many effectors have been identified for some well-studied Rabs, such as Rab5, there other Rabs where many fewer effectors have been found, and even for Rab5 we were able to find and validate new effectors. As for GEFs we acknowledge that we were only able to identify known GEFs for 4 of the 11 Rabs tested. However, for the Rho and Ras family GTPases we detected known GEFs as strong hits for five of the six we tested, and the one exception, Rheb, has no known GEF. Thus, of the seventeen GTPases tested, nine identified known GEFs, which seems better than “mostly ineffective”.

As for why we did not detect GEFs for all of the Rabs, we note in the Discussion that the GEFs are thought to preferentially bind the empty form of the GTPase and it may be that the amount of empty form present for the “GDP-locked” mutant is somewhat variable between different GTPases. In addition, the empty form may in some cases be unstable, and indeed the protein blotting we have added to the paper reveals that three of the GTPases did not express well in the GDP-locked form, an observation previously reported for Rab27A. It is possible that a different mutation in the nucleotide binding pocket would work better, but seeking such mutations seems beyond the scope of this study, and so we have instead noted this as a possible route forward.

Finally, the proteins identified with Rab2::GDP and Rab11::GDP do not include strong candidates for potential novel GEFs. In both cases, all known proven or putative GEFs are identified, and the few other proteins not only have known functions but are less strong hits and are also hits with the GTP-bound state. This probably reflects the fact that a small number of effectors and some GAPs bind to the GTPase independently of nucleotide state. For instance, the π 4-kinase PI4KB is a hit with Rab11::GDP but it is also a hit with Rab11:GTP consistent with previous biochemical and structural work showing that Rab11 binds to PI4KB in a nucleotide-independent manner (Burke et al., 2014). We apologise for not making this important point clearer in the text, and have now added text to the Results section and the legend to Figure 6 noting the existence of such proteins.